



# Determination limits for cosmogenic [10]Be and their importance for geomorphic applications

Sara Savi[1], Stefanie Tofelde[1,2], Hella Wittmann[2], Fabiana Castino[1] and Taylor F. Schildgen[1,2]

[1] Institute of Earth and Environmental Science, University of Potsdam, Karl-Liebknecht-Str. 24, Haus 27, 14476 Potsdam-Golm, Germany
[2] Helmholtz Centre Potsdam, GFZ German Research Centre for Geosciences, Telegrafenberg, 14473 Potsdam, Germany

*Correspondence to*: Sara Savi (sara.savi@geo.uni-potsdam.de)



**Abstract.** When using cosmogenic nuclides to determine exposure ages or denudation rates in rapidly evolving landscapes, challenges arise related to the small number of nuclides that have accumulated in surface materials. Improvements in accelerator mass spectrometry have enabled analysis of samples with low $^{10}$Be content ($<10^5$ atoms), such that it is timely to discuss how technical limits of nuclide determination, effects of laboratory cleanliness, and overall sample preparation quality affect lower blank limits. Here we describe an approach that defines a lower threshold above which samples with low $^{10}$Be content can be statistically distinguished from laboratory blanks. In general, this threshold depends on the chosen confidence interval. In detail, however, we show that depending on which ensemble of blank values and which approach is chosen for the calculation of this threshold, significant differences can arise with respect to when a sample can be distinguished from a blank. This in turn dictates whether the sample can be used to determine an exposure age or a denudation rate, or when it only constrains a maximum age or a minimum denudation rate. Based on a dataset of 57 samples and 61 blank measurements obtained in one laboratory, we demonstrate how these different approaches may influence the interpretation of the data.





## 1 Introduction

In the last two decades, the use of *in situ*-produced cosmogenic nuclides for the quantification of denudation processes and the determination of exposure ages of landforms has seen a rapid expansion (Balco, 2011; Granger et al., 2013). This development is due to advances in the technique and to the wide range of geological environments in which the method can be applied. Comprehensive summaries of the method can be found in Anderson et al. (1996), Bierman and Steig (1996), Granger et al. (1996), von Blanckenburg (2005), Balco (2011), and Granger et al. (2013). Among the suite of cosmogenic nuclides that can be used to study geomorphic processes (e.g., $^{10}$Be, $^{26}$Al, $^{36}$Cl, $^{3}$He, and $^{21}$Ne), *in situ*-produced $^{10}$Be is the most widely used, especially for the quantification of denudation rates (von Blanckenburg, 2005). For simplicity and clarity, we will focus our discussion on *in situ* $^{10}$Be produced in the target mineral quartz only, although similar concepts can be applied to other cosmogenic nuclides. The broad expansion of $^{10}$Be applications includes studies that extend the limits of the technique by analyzing nuclide concentrations in environments where some of the assumptions inherent to the method are not always satisfied. These studies explore, for example, landscapes that are not in erosional steady-state, e.g. due to recent glaciation (Wittmann et al., 2007), settings where different rock types do not contribute quartz equally (Safran et al., 2005; Torres Acosta et al., 2015), and environments prone to mass failures or with non-uniform sediment supply (Niemi et al., 2005; Binnie et al., 2006; Yanites et al., 2009; Kober et al., 2012; McPhillips et al., 2014; Savi et al., 2014; Schildgen et al., 2016). It is particularly challenging to apply these techniques in environments where the cosmogenic nuclide content is low. For example, the occurrence of deep-seated landslide or debris-flow events in rapidly eroding landscapes may result in admixing of low cosmogenic nuclide concentration material into fluvial sediments (Niemi et al., 2005; Yanites et al., 2009; Kober et al., 2012; Savi et al., 2014). Likewise, recently exposed bedrock surfaces contain low $^{10}$Be due to their short exposure time to cosmic rays (Licciardi et al., 2009; Schaefer et al., 2009; Schimmelpfenning et al., 2014; Savi et al., 2016). In other cases, scarcity of the target mineral within the collected material can limit the total amount of nuclides in the sample. Difficulties encountered with low $^{10}$Be-content samples are related to the technical limits of the Accelerator Mass Spectrometer (AMS), which can precisely measure $^{10}$Be/$^{9}$Be ratios down to $10^{-15}$ or $10^{-16}$ (Stone, 1998), as well as technical limits and cleanliness issues related to the laboratory where the samples are prepared (e.g., Balco, 2011).

Many AMS facilities provide the lower limit at which they can precisely measure nuclide ratios (e.g., Rood et al., 2010). However, this number does not account for laboratory cleanliness and contamination that may be introduced during sample preparation and chemical procedures, which are reflected in laboratory blanks. Nevertheless, several recent studies demonstrated that it is possible to date very young landforms (such as moraines of the Little Ice Age) if a series of strict sampling and laboratory standards (including low laboratory blank measurements) are met (Licciardi et al., 2009; Schaefer et al., 2009; Schimmelpfenning et al., 2014). As such, the adoption of a standardized procedure for defining measurement thresholds is timely and valuable in the field of geomorphic studies. Despite the existence of a procedure commonly used in analytical chemistry to define a statistically significant threshold for low-concentration samples (e.g., Long and Winefordner, 1983), there have been so far no clear guidelines on how to apply this procedure to cosmogenic nuclide data.

In this paper, we describe this method and evaluate its application for $^{10}$Be studies. The method is based on laboratory blank measurements, thereby not only accounting for the AMS detection limit, but also for the





cleanliness and contamination that may occur in the laboratory. As an example, we use the [10]Be measurements of 57 samples and 61 laboratory blanks to address (1) the determination of statistically significant limits that define the lower threshold for quantifying exposure ages or denudation rates, and (2) the influence of the approach used for the calculation of this threshold on the quantitative use of the samples.

**2 State of the Art**

**2.1 Cosmogenic nuclide techniques in Earth Surface Sciences**

*In situ* cosmogenic nuclides are produced when secondary cosmic rays (formed by nuclear reactions in the atmosphere) collide with target minerals at or near the Earth's surface (Nishiizumi et al., 1989; Lal, 1991; Brown, 1992; von Blanckenburg, 2005; Dunai, 2010). Being a rare cosmogenic radioactive nuclide with a half-life of 1.39

Myr (Chmeleff et al., 2010; Korschinek et al., 2010), [10]Be is not naturally present in Earth surface materials. Because the production rate of [10]Be decreases approximately exponentially with depth, nuclide accumulation mostly occurs within the upper few meters of the surface (Lal, 1991; Dunai, 2010). For this reason, [10]Be is widely used for two main categories of geomorphological research: (1) dating of exposed or buried surfaces and sediment (e.g., Ivy-Ochs and Kober, 2008; Granger, 2006) and (2) quantification of recent (typically millennial-scale)

denudation rates (e.g. von Blanckenburg, 2005; Granger et al., 2013). In these applications, the [10]Be content is measured in a sample collected from an exposed surface (e.g., bedrock or a large boulder) or shielded deposit, or from a sample of sediment collected from a river-bed or sedimentary deposit (Dunai et al., 2000; Gosse and Phillips, 2001; von Blanckenburg, 2005; Nishiizumi et al. 2007; Dunai, 2010). The [10]Be content of a sample is directly related to the time that the sample has been exposed to cosmic rays (minus those lost due to decay) and is

inversely related to the erosion rate (Nishiizumi et al., 1989; Lal, 1991). Hence, the [10]Be content of a sample depends on both the nuclide´s concentration and the mass of the target mineral (e.g., quartz) from which [10]Be is extracted.

**2.2 From AMS ratios to denudation rates and exposure ages**

Once Be is extracted from the sample, the [10]Be (and [9]Be) atoms contained in a target mineral are detected with

an AMS. This instrument uses a source of accelerated ions and a series of magnets to separate different chemical elements with similar atomic masses (e.g. Dewald et al., 2013), which can be interpreted in terms of a [10]Be/[9]Be ratio based on a known measurement standard (Rugel et al., 2016). To calculate the number of [10]Be atoms in a sample, the derived [10]Be/[9]Be ratio is multiplied by the amount of [9]Be in the sample, which is normally the known amount added as [9]Be carrier during sample preparation. Most rock types contain the trace metal [9]Be in the lower

ppm-range (Rudnick and Gao, 2004), which makes the total contribution of natural [9]Be negligible relative to the amount added by the carrier. Recent studies, however, have demonstrated that in some rock types the natural [9]Be is high enough that it must be quantified and included when converting [10]Be/[9]Be ratios to [10]Be content (Portenga et al., 2015; Corbett et al., 2016).

Common chemical preparation procedures include the processing of one or more procedural blanks (referred

to as *laboratory blanks* hereafter), containing the [9]Be carrier only, to help quantify any contamination that may





occur during sample preparation and to account for any $^{10}$Be within the carrier. To obtain the number of $^{10}$Be atoms in the sample related to *in-situ* production, a *blank correction* or *blank subtraction* is commonly performed.

### 2.3 Blank types and blank corrections

*Laboratory blanks* are typically introduced during the Be separation stage right before total sample dissolution. Samples are likely to undergo different preparation histories (e.g., different amounts of acid used for dissolution, different time-spans needed for evaporation, etc.) that could result in differing amounts of $^{10}$Be contamination (Balco, 2011), which are reflected in the laboratory blanks (see *supplementary material, text S5 and S6)*. In addition, the amount of $^{10}$Be contamination introduced with the $^{9}$Be carrier from commercial Be solutions is not negligible (Balco, 2011; Granger et al., 2013). Merchel et al. (2008) reported $^{10}$Be/$^{9}$Be ratios between $10^{-14}$ to $10^{-15}$ for a range of commercially available Be solutions that may be used as carrier. Deep-mined phenakite or beryl minerals can alternatively be used to produce carrier solutions, in which case $^{10}$Be/$^{9}$Be ratios in the range of $10^{-16}$ have been obtained (e.g. Stone, 1998; Schaefer et al., 2009; Merchel et al., 2013; Portenga et al., 2015; Corbett et al., 2016).

*Machine* or *instrument blanks* (generally related to AMS measurements) indicate the precision at which the AMS can measure the $^{10}$Be/$^{9}$Be ratio (Balco, 2011). This latter kind of blank is uninfluenced by contamination that occurs during the chemical procedure in the laboratory and can provide information about the sensitivity of the measuring process. In accelerator mass spectrometry, cross-contamination due to long-term memory of the AMS measurements is in the order of 0.1‰ (Rugel et al., 2016), so that the machine background is commonly neglected (Currie, 2008). It follows that most of the contamination stems from the laboratory processing of the samples (Balco, 2011) and thus we focus only on *laboratory blanks* hereafter.

### 2.3.1 Blank corrections

To perform a *blank correction*, the number of $^{10}$Be atoms contained in the blank is subtracted from the number of $^{10}$Be atoms contained in the sample. When the number of $^{10}$Be atoms in a sample is significantly greater than in the blanks, the value and variability of multiple blank measurements has little impact on the blank-corrected result. However, in the case of low $^{10}$Be content in a sample, the blank correction constitutes a large subtraction and variations among individual blanks become important.

Some $^{10}$Be in the blank may originate from processes that would affect an entire batch of samples that are processed together, such as the $^{10}$Be contained in the carrier or in the stock chemicals used (see *supplementary material, text S5)*. In these cases, a single blank per batch probably provides a good measure of the $^{10}$Be contamination. Other sources, such as cross-contamination from poor laboratory practices or insufficient cleaning of reusable labware result in variable contamination among samples of the same batch and using a single blank per batch may thus be inadequate. Bierman et al. (2002) provide details of replicate blank measurements from 53 batches of samples processed at the cosmogenic-nuclide target preparation laboratories of the University of Vermont, where two blanks were processed per batch. The good agreement between these blank pairs suggests, at least in that laboratory, that inter-batch contamination is not an issue. However, this is a point that needs to be addressed for each laboratory and perhaps at the individual-user level.





### 2.4 Determination limits

#### 2.4.1 General statistical background

Following the *International Union of Pure and Applied Chemistry* (IUPAC) definition, there is a minimum sample concentration that can be determined to be statistically different from an analytical blank in every analytical
procedure (Long and Winefordner, 1983). The term "statistically different" implies the application of a statistical approach that tries to answer the question "what is the lowest sample concentration that can be reliably distinguished from a blank?" (Currie, 1968; Long and Winefordner, 1983; McKillup and Darby Dyar, 2010; Schrivastava and Gupta, 2011; Bernal, 2014). This question can also be formulated as "what is the upper value of the blank distribution (i.e. the distribution of all available $^{10}$Be blank measurements) that ensures a reliable
distinction between blank and sample amounts?"

For cosmogenic studies, the previous question can be translated into the following *null hypothesis* (Fig. 1): "The number of $^{10}$Be atoms in a given sample is not distinguishable from that within the blank(s)", which must be tested at a fixed confidence interval. Here, we give the example of a 1-tail test, because we are interested in defining an upper limit for the blank distribution. For variables that are normally distributed, the most common values used
for confidence intervals, calculated according to equation (1) below, are $\pm k\sigma$, with $k = 1$, 2, or 3 (McKillup and Darby Dyar, 2010), and $\sigma$ being the standard deviation of the distribution. Whenever the lower limit of the sample's $^{10}$Be value, including the uncertainty interval, lies outside of a chosen confidence interval, the null hypothesis can be rejected, and we can infer that the sample is statistically distinguishable from the blank (Fig. 1). It follows that the choice of the confidence interval to use for the definition of a statistically distinguishable value defines the
determination limit (McKillup and Darby Dyar, 2010) and, consequently, if a sample can be distinguished from the blanks (Fig. 1).

#### 2.4.2 Limits of detection (LOD) and quantification (LOQ)

The IUPAC and the ACS (*American Chemical Society*) recommend, by generalizing the formulas first introduced by Currie in 1968, calculating the limits of determination with the following equation:
$$Limit = \mu_{Blk} + k\sigma_{Blk} \tag{1}$$

where $\mu_{Blk}$ and $\sigma_{Blk}$ are the mean and standard deviation of a normal distribution of all blank measurements (Analytical Methods Committee, 1987), which ideally contains at least 20 individual blank measurements (Bernal, 2014). When k=3, Eq. (1) defines the LOD (Limit of Detection), known as "*the lowest concentration, or amount, of an analyte that can be detected with reasonable confidence for a given analytical procedure*" (Analytical
Methods Committee, 1987; Mocak et al., 1997), according to:
$$LOD = \mu_{Blk} + 3\sigma_{Blk}. \tag{2}$$

This definition of the LOD fixes the confidence interval for the blank distribution, assumed to be Gaussian, at $3\sigma$ (99.9% for 1-tail test). Alternatively, when k=10 in Eq. (1), the LOQ (Limit of Quantification) is defined as the lower limit above which every analytical sample can be used in a quantitative way with a confidence interval larger
than 99.9999% (Analytical Methods Committee, 1987):
$$LOQ = \mu_{Blk} + 10\sigma_{Blk}. \tag{3}$$



From these definitions of the LOD and LOQ, it is clear that the variability of the blank measurements (expressed through the variance, $\sigma^2$) has a large influence on the final value of the limits. In particular, the LOD and LOQ will be relatively close to the mean value of the blank distribution when it has a low variance, whereas the LOD and LOQ will be substantially larger than the mean when the blank distribution has pronounced variance.

**3 Method and study design**

To be able to statistically distinguish a sample with low $^{10}$Be content from a blank, a proper confidence interval needs to be chosen and related to the distribution of the laboratory blanks. In 1987, the Analytical Method Committee suggested to follow the IUPAC recommendations (i.e., equations (2) and (3)). This recommendation, however, assumes a normal distribution of values, which is rarely the case when dealing with low concentrations

of an analyte (Currie, 1972; Bernal, 2014). As such, following this assumption could result in slightly different confidence intervals for the calculated determination limits, as we will illustrate later in Sect. 4. Importantly, if the number of blanks used for the calculation of the determination limits is low (e.g., less than 20), or if the distribution of the measured blank values is not normal, the use of $\mu$ and $\sigma$ for the estimation of the LOD and LOQ defined by the IUPAC's equations may be not appropriate (Long and Winefordner, 1983; Analytical Methods Committee,

1987; Currie, 1968; 1972; Bernal, 2014).

In the case of a low number of blank measurements, which may not accurately constrain laboratory contamination, we overcome this limitation by including additional blank measurements collected over a long time period from the same laboratory. However, this approach assumes that the whole laboratory process is unchanging and that long-term variations in the blanks are representative of the variation within a single batch. In case of non-

normal distributions, to estimate the determination limits at fixed confidence intervals we use an alternative approach to Eq. (2) and (3). This approach entails (a) identifying which distribution best describes the measured blank data, and (b) calculating the percentile of that distribution relative to the chosen confidence interval. For example, if the best-fit to the blank data is represented by a *Negative Binomial* distribution, fixing the confidence interval at 99.9% (equivalent to the LOD) would require that only samples with a number of $^{10}$Be atoms above the

99.9$^{th}$ percentile of the blank distribution can be considered to be significantly different from the blanks. As such, once having determined the distribution that best describes the blank ensemble, the determination limits are given by the percentiles equivalent to the desired confidence level.

**3.1 Dataset**

To test these two approaches (IUPAC recommendations versus distribution percentiles) and evaluate their

implications for geomorphic studies, we use one set of 61 blank values (Table 1) and one set of 57 sample values (35 of which are published in Savi et al., 2016) (Table 2). Blanks and samples were prepared at the Helmholtz Laboratory of the Geochemistry of Earth Surface (HELGES) at GFZ Potsdam (von Blanckenburg et al., 2016) between July 2013 and May 2016. The set of 61 blanks includes eight blanks simultaneously processed with the batches of samples over a few 1-month periods between 2014 and 2016. These eight blanks were processed by the

same operator, whereas the long-term blanks were prepared by different users. After having converted $^{10}$Be/$^9$Be ratios from the AMS into the number of $^{10}$Be atoms, we: (1) calculate the LOD and LOQ values for both normal





and negative binomial blank distributions associated with both the eight-blank ensemble and the 61-blank ensemble; (2) assess the reliability of each single sample by comparing its [10]Be content to the LOD and LOQ values; and (3) perform different blank corrections to obtain the final [10]Be content of our samples.

### 3.2 Determination limits (LOD and LOQ) for cosmogenic [10]Be

Although a minimum of 20 values is generally needed for a statistically significant description of a data ensemble (Long and Winefordner, 1983; Analytical Methods Committee, 1987; Bernal, 2014), for comparison purposes, we determined the LOD and LOQ values for two sets of blanks: (1) the eight blanks processed together with the samples by one operator, and (2) the 61 long-term blank values processed in the laboratory by multiple users. In this way, we are able to explore a real situation and observe the effects of the different approaches on the

calculation of the LOD and LOQ values. We are aware that the use of the LOQ as a lower threshold may be overly restrictive for cosmogenic studies, because such a conservative confidence interval (i.e., 99.9999%) is commonly not needed in this field. However, for the completeness of this study, we discuss all the different results and their implications in Sect. 4 and 5.

     For both blank datasets, we calculated the LOD and LOQ using Eq. (2) and (3) assuming a Gaussian

distribution, as well as by fitting a negative binomial curve to the blank distribution using the Matlab[TM] *Distribution Fitting* toolbox (within the Statistic and Machine Learning toolboxes). The latter requires calculating percentiles, which can be performed using the same toolbox. With this approach, we can compare the effects of having different blank distributions (Fig. 2). To assess whether or not a sample falls below the calculated limits, we compare the lower number of [10]Be atoms contained in the sample (i.e., the number of uncorrected [10]Be atoms

in a sample minus its uncertainty, generally expressed by 1σ) (Table 2) to the various LOD and LOQ values.

### 3.3 Blank correction methods

     Samples were corrected for blanks after having assessed which samples can be reliably distinguished from the blanks. Based on the data ensemble and study design mentioned in Sect. 3.1, three different procedures for the blank correction can be applied: (1) a *single-batch blank correction*, (2) an *average-blank correction*, and (3) a

*long-term laboratory blank correction*.

     For the *single-batch blank correction*, the number of [10]Be atoms calculated from a single blank processed along with the sample batch is subtracted from each sample within the batch. In this case, the AMS uncertainties associated with each sample and the individual blank measurement are then used for error propagation (Peters, 2001). In the *average-blank correction*, all the blanks processed in multiple batches by one operator over a limited

time frame (i.e., the eight blanks used for the calculation of the determination limits) are used to obtain a representative value of [10]Be atoms for the blanks (e.g., the value that best describes the blank distribution – see below). This representative value is then subtracted from all samples. In contrast, the *long-term laboratory blank correction* subtracts a [10]Be value obtained from all the blanks measured at the laboratory over a longer time span from each sample (this value is obtained from the 61 blank values from which the determination limits are

calculated). As such, the *average blank correction* accounts for operator-specific abilities, any temporal component of the laboratory background, and the variability that may occur among different sample batches. The *long-term blank correction* method has the advantage of being based on many measurements (and hence is





statistically well constrained). However, because it includes variability associated with different operators and potential variations among batches, it will likely overestimate the variability associated with a single operator processing a single batch (or a limited number of batches) of samples.

For the *average-blank correction* and the *long-term laboratory blank correction* methods, the representative

value of $^{10}$Be atoms used for the blank subtraction can either be the mean of the blank distribution, or its median, depending on which parameter best characterizes the blank ensemble. The uncertainty associated with the chosen representative value that is then used for error propagation procedures is given by the standard error of the mean ($\varepsilon_\mu$) or the standard error of the median ($\varepsilon_m$) of the blank frequency distribution, calculated as follows (Evans, 1942; Peters, 2001):

$$\varepsilon_\mu = \frac{\sigma}{\sqrt{n}} \qquad (4)$$

$$\varepsilon_m = \frac{C\sqrt{n}}{2f} \qquad (5)$$

where $\sigma$ is the standard deviation of the blank ensemble, $n$ is the number of blank measurements in the blank distribution, $C$ is the width of the bins used for the blank distribution, and $f$ is the number of blanks falling in the $C$-bin that includes the median (Figure S2 and Table S2).

## 4 Results

### 4.1 Determination limits

The distribution of the eight blanks is characterized by a mean ($\mu$) of 0.76 x 10$^4$ $^{10}$Be atoms, a standard deviation ($\sigma$) of 0.39 x 10$^4$ $^{10}$Be atoms, and a median of 0.70 x 10$^4$ $^{10}$Be atoms. The standard error of the mean is 0.14 x 10$^4$

$^{10}$Be atoms and the standard error of the median is 0.09 x 10$^4$ $^{10}$Be atoms (for C=2000 and f=3 in Eq. (5)). The 61 long-term blanks are characterized by a mean of 1.72 x 10$^4$ $^{10}$Be atoms, a standard deviation of 1.96 x 10$^4$ $^{10}$Be atoms, and a median of 0.93 x 10$^4$ $^{10}$Be atoms. The standard error of the mean is 0.25 x 10$^4$ $^{10}$Be atoms and the standard error of the median is 0.16 x 10$^4$ $^{10}$Be atoms (for C=2000 and f=5 in Eq. (5)).

For both sets of blank values (subscript "8" for the eight blanks and subscript "61" for the 61 long-term blanks),

the LOD and LOQ values calculated following the IUPAC recommendations (subscript "N", assuming the blanks are normally distributed) are always smaller than the ones calculated using the distribution percentiles (subscript "NB", assuming the blanks are best described by a negative binomial distribution) (Table 3, Fig. 3). In particular, for the long-term blank ensemble, which is non-normal and strongly positively skewed (Fig. 2), the $LOD_{61,N}$ calculated with Eq. (2) corresponds to a confidence interval of 98.9% of the actual distribution rather than 99.9%.

The same is true for the eight-blank ensemble, where the $LOD_{8,N}$ estimated using Eq. (2) corresponds to a confidence interval of 99.3% (rather than 99.9%) because the values are non-normally distributed (Fig. 2).





### 4.2 Reliability of the samples

The number of samples falling below the different determination limits (Table 3, Fig. 3) varies depending on which blank ensemble is chosen and which approach is used to calculate the determination limit (i.e., IUPAC recommendations versus distribution percentiles). For the eight blanks processed by one operator, the approach

used to calculate the determination limits does not significantly influence their values nor the number of samples falling below these thresholds (Fig. 3). In contrast, for the 61 long-term laboratory blanks, the approach used for the calculation of the limits has a strong impact on the values of the limits and consequently also on the number of samples that can be quantitatively used to calculate exposure ages or denudation rates (Table 3, Fig. 3).

### 4.3 Blank corrections and error propagation

Not surprisingly, for samples with low $^{10}$Be content (i.e., $< 10^6$ $^{10}$Be atoms), the corrected $^{10}$Be values vary depending on which blank correction method is applied. Considering only the samples that can be reliably distinguished from the blanks (i.e., above LOD values), the *single-batch blank correction* commonly yields the highest sample $^{10}$Be content. These values, however, are not always associated with the lowest uncertainty, which is in some cases given by the *average blank correction* method (Table S2). The maximum difference in $^{10}$Be

content between these two correction methods is associated with the sample with the lowest $^{10}$Be content, and amounts to 13.6%. Within the *average blank correction* method, the choice of the mean (0.76 x $10^4$ $^{10}$Be atoms) or median (0.70 x $10^4$ $^{10}$Be atoms) as the representative value for the blank subtraction has little influence on the final $^{10}$Be content of the samples (maximum difference $< 2.5\%$). Also, the difference in the propagated uncertainties among samples is negligible (Table S2).

In general, samples corrected with the *single-batch blank correction* and the *average blank correction* methods yield higher values with lower uncertainties than the ones calculated with the *long-term blank correction* method. When this latter correction method is applied, the choice of the mean (1.72 x $10^4$ $^{10}$Be atoms) or median (0.93 x $10^4$ $^{10}$Be atoms) as the representative value for the blank subtraction yields slightly different results. The maximum difference in $^{10}$Be content obtained when using the mean or median is associated with the sample with the lowest

$^{10}$Be content, and amounts to 6.5%. Also in this case, the difference between the propagated uncertainties among samples is negligible (Table S2).

### 5 Discussion

Our results highlight that whether or not a sample measurement lies above the chosen determination limit depends on: (1) the distribution of blanks, (2) the type of applied determination limit (LOD or LOQ), and (3) the

approach used for its calculation (i.e., IUPAC recommendations versus distribution percentiles). All samples for which the $^{10}$Be content is lower than the chosen determination limit cannot be considered as reliably distinguishable from the laboratory blanks. Therefore, these values can only be used to limit the age (i.e., younger than the youngest detectable age) or denudation rate (i.e., faster than the fastest detectable rate) recorded by a sample. In this sense, the LOD or LOQ value can be considered as the youngest detectable age or fastest detectable

rate measurable from a data ensemble.





### 5.1 Carrier quality, blank distribution, and determination limits

The results of our analysis highlight that the choice of the blank ensemble from which to calculate the determination limits is among the most important factors in defining the final results and the number of samples that can be statistically distinguished from the blanks.

For the eight-blank ensemble, blank values are relatively low and show low variance, which leads to low corresponding $LOD_8$ and $LOQ_8$ values. It follows that most of the samples lie above the determination limits (although 20% of samples cannot be distinguished from blanks when the $LOQ_8$ is chosen as the threshold). These determination limits, being based on only eight blanks, are representative of the individual operator's ability and the contamination that occurred in the laboratory during sample processing. Compared to the eight-blank

ensemble, the long-term laboratory blanks have larger variance and show values spanning two orders of magnitude. As such, the relatively high $LOD_{61}$ and $LOQ_{61}$ values imply that at least 16 samples, and up to 29 samples cannot be reliably distinguished from the blanks (Fig. 3, Table 3).

In general, the use of the long-term laboratory blanks (being based on many blank measurements) guarantees more reliable values for the statistics of the blank distribution and for the calculation of the determination limits;

as such, they may be preferred. Nevertheless, when the long-term blank ensemble shows a large variance, the assumption of unchanging laboratory conditions is unlikely to be valid, and the blank measurements are unlikely to be representative of the variation occurring within a single batch. Under these circumstances, and when there is an acceptable number of blank measurements available (at least 20; Bernal, 2014), a set of blanks obtained from a single operator over a shorter time interval may be favoured for the calculation of the threshold. As such, and

especially when samples with low $^{10}$Be content are expected, processing more than one blank per batch (Fig. 4) guarantees better constrained (i.e., statistically based on a larger number of measurements) determination limits. It is important to remark that choosing a subset of blanks for the calculation of the determination limits does not reflect the long-term history of the laboratory. This approach can only be used to limit the ages or erosion rates of the samples processed in the same time-span of when the subset of blanks was processed. Also, in the case of low-

concentration samples, the effort and/or expense of acquiring low $^{10}$Be/$^9$Be carrier, which typically defines the lower attainable limit for the laboratory blank measurements, is most likely worthwhile. When estimating the determination limits, it is also important to consider whether to follow the IUPAC recommendations (i.e., assuming a Gaussian distribution), or to use the distribution percentiles for a more accurate estimation of the determination limits (Fig. 4).

### 5.2 Increasing the precision of AMS measurements

In general, samples with larger uncertainties have a higher chance of falling below the determination limit. Accordingly, apart from improvements in laboratory cleanliness, another way to decrease the uncertainty on the $^{10}$Be content of a sample would be to increase the precision of the AMS measurement. This could be done by increasing the amount of sample processed during the chemical procedure, by reducing the loss of quartz during

the chemical treatment of the samples (see *supplementary material, text S4*), or by increasing the AMS current during measurement (Schaefer et al., 2009; Rugel et al., 2016; Fig. 4). Whether to increase the amount of sample to process, or to use specific techniques to minimize quartz loss during Be-purification may depend on the laboratory standards, costs, processing-times, and the feasibility of acquiring large amounts of sample.





### 5.3 Which limit do I use?

Given the high confidence intervals that are often necessary in particular applications of chemistry, such as when dealing with contaminants (McKillup and Darby Dyar, 2010), the IUPAC and ACS have defined the LOQ as the limit for the quantitative use of the sample data. However, for cosmogenic studies, there is little need for

such high confidence intervals and consequently conservative interpretations. As such, the choice of the LOD as the lower threshold is reasonable, and still allows users to distinguish samples from blanks with a reliability of 99.9%. When such high confidence is not required, the users may alternatively use a $2\sigma$ confidence interval (i.e., equal to 97.7% for 1-tail test) as suggested by Stuiver and Polach (1977) for radiocarbon measurements. Accordingly, the $LOD_{97.7}$ is calculated with $k = 2$ in Eq. (1) if the blank distribution is Gaussian, or using the

97.70th percentile in the case of a non-normal blank distribution.

Once calculated, the LOD represents the lowest number of $^{10}Be$ atoms that can be distinguished from the blanks and, thus, can be used to limit the ages or the denudation rates shown by the dataset. Our results indicate that, after having established which samples are above the defined threshold, the type of blank correction and the representative value used for the blank subtraction (i.e., mean versus median value), may be important for samples

with low $^{10}Be$ content, whereas their importance becomes negligible for samples with more than $10^6$ $^{10}Be$ atoms (Fig. 5). However, because the LOD and the results of the blank correction are strongly dependent on the chosen blank ensemble, the final decision on which approach to use is best evaluated case by case (Fig. 4).

### 5.4 Implications for geomorphic applications

With our blank and sample datasets, we demonstrated that depending on which approach is used for the

calculation of the determination limits, the number of samples that can reliably be distinguished from blanks may vary strongly. In particular, considering only the LOD values for both subsets of blanks, we showed that the percentage of samples that cannot be reliably distinguished from the blanks varies between 5% and 37% (Fig. 3). This range shows that for geomorphic applications, the precision of the measurements and the cleanliness of the laboratory procedure can have a strong impact on the final interpretation of the data. For example, by using low

$^{10}Be/^9Be$-ratio carrier and increasing the AMS current, Schaefer et al. (2009) were able to obtain very low and precise blank values (between 6,000 and 26,000 atoms with $1\sigma < 10\%$) and very precise sample measurements (between 70,000 and 1,000,000 atoms with $1\sigma < 10\%$). When applying the IUPAC recommendations to this dataset, the LOD is around 58,500 $^{10}Be$ atoms, implying that all the samples are statistically distinguishable from the blanks and can be used with a confidence of 99.9%. With these highly precise results, the authors measured

exposure ages as young as $150 \pm 15$ years, dating moraines of the Little Ice Age.

In a case of similarly young boulders and rapidly denuding tributary catchments on an alluvial fan, Savi et al. (2016) showed that depending on which blank correction method is applied, the propagated error on the final exposure ages or denudation rates can vary up to 20%. Also, considering the $LOD_N$ as the lower threshold, these authors had 3 samples (about one fifth of a first set of processed samples) that could not be distinguished from the

blanks, and could only be used to limit the exposure ages or denudation rates. However, by increasing the amount of sample processed during the chemical procedure for the following set of samples, and thus increasing the precision of the AMS measurements, all of the remaining samples could be statistically distinguished from the blanks with a confidence of 99.9%. These results were interpreted as quantifiable exposure ages as low as $50 \pm 8$



years, and denudation rates as fast as 13.8 ± 2.6 mm/yr. This study highlights how it may be possible to measure young exposure ages and fast denudation rates at a very high confidence level.

**6 Conclusions**

In this paper we have discussed the challenges related to the use of cosmogenic nuclide techniques in the case

of low $^{10}$Be content, which are typically found in samples collected from rapidly eroding landscapes, young surfaces, or when a very limited amount of the target mineral is available for analysis. By adapting a method commonly used in analytical chemistry, we describe an approach to define a lower threshold above which samples with low $^{10}$Be content can be used in a quantitative way, accounting for laboratory cleanliness and contamination that may occur during the chemical procedure. This approach can be applied by the end-user of AMS

measurements based on a number of different options for characterizing laboratory blanks.

In summary, in an ideal situation, the use of at least 20 blank values would guarantee a statistically reliable value for the limit of detection, LOD, which can be considered as the lowest threshold for the quantitative use of cosmogenic nuclide data. When samples with low $^{10}$Be content are expected, the user can process multiple blanks within a single sample batch. As an alternative, one can use a long-term value derived from several laboratory

blanks processed over a limited time frame during which laboratory conditions are assumed to be nearly constant. When an acceptable number of blank values is available (i.e., minimum 20), finding the distribution that best describes the blank ensemble and using the 99.9$^{th}$ percentile of that distribution for the calculation of the LOD guarantees a more precise estimate of this threshold at the fixed confidence level.

Our analysis demonstrates the importance of producing low, precise, and reproducible blank measurements, as

they reduce the value of the various determination limits, therefore increasing the number of samples that are distinguishable from laboratory blanks. Particularly for samples with low $^{10}$Be content, it is important to report detailed information about the laboratory protocols, blank measurements (including both the measured ratios and the amount of added carrier, in order to calculate the number of $^{10}$Be atoms in the blanks), the value and the procedure used to calculate the chosen determination limit, and the applied blank correction method.



**Author Contribution**

All authors agreed and contributed to the article. SS, ST, HW, and TS developed the design of this work; SS prepared the samples and did the chemical procedure, FC supported the team in the statistic section and all the co-authors supported the first author in writing the text.

5 **Competing Interests**

The authors declare that they have no conflict of interest.

**Acknowledgments**

This work has been funded by the Swiss National Science Foundation (grants PBBEP2_146196 and P300P2_151344 awarded to S. Savi), the Alexander von Humboldt Foundation (grant ITA 1154030 STP awarded 10  to S. Savi) and the Emmy Noether program of the German Science Foundation (grant SCHI 1241/1-1, awarded to T. Schildgen). We greatly acknowledge the helpful and detailed discussions with S. A. Binnie of the University of Cologne and the detailed comments of Lee Corbett and of an anonymous reviewer on an earlier version of this manuscript.





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



**Table 1. Long-term blank measurements at the German Research Center (GFZ) from July 2013 to May 2016, including the 8 blanks (in bold) processed within the samples' batches. A carrier addition of ca. 0.15 mg $^9$Be (in accordance with the laboratory protocols) was used for the calculation of the $^{10}$Be atoms.**

| Number of Blanks | $^{10}Be/^9Be$ | $^{10}Be$ atoms from carrier | $1\sigma$ [%] | Number of Blanks | $^{10}Be/^9Be$ | $^{10}Be$ atoms from carrier | $1\sigma$ [%] |
|---|---|---|---|---|---|---|---|
| 1 | 5.96E-16 | 5.97E+03 | 50.1 | 32 | 1.68E-15 | 1.68E+04 | 26.9 |
| 2 | 3.14E-16 | 3.15E+03 | 70.8 | 33 | 1.00E-15 | 1.00E+04 | 71.1 |
| 3 | 6.11E-16 | 6.12E+03 | 44.8 | 34 | 5.62E-16 | 5.63E+03 | 70.6 |
| 4 | 9.86E-17 | 9.88E+02 | 100.0 | 35 | 5.21E-16 | 5.23E+03 | 50.1 |
| 5 | 1.08E-15 | 1.08E+04 | 44.8 | 36 | 6.23E-15 | 6.24E+04 | 12.7 |
| 6 | 3.05E-16 | 3.06E+03 | 70.8 | 37 | 7.30E-15 | 7.32E+04 | 12.0 |
| 7 | 1.05E-15 | 1.05E+04 | 18.4 | **38** | **6.90E-16** | **6.99E+03** | **44.8** |
| 8 | 1.94E-15 | 1.94E+04 | 13.6 | 39 | 6.84E-15 | 6.85E+04 | 15.9 |
| 9 | 1.00E-16 | 1.00E+03 | 100.0 | 40 | 3.06E-15 | 3.07E+04 | 19.5 |
| **10** | **2.48E-16** | **2.52E+03** | **70.8** | 41 | 1.95E-15 | 1.96E+04 | 23.1 |
| **11** | **4.24E-16** | **4.29E+03** | **57.8** | 42 | 9.65E-15 | 9.68E+04 | 11.6 |
| **12** | **7.27E-16** | **7.34E+03** | **40.9** | 43 | 6.58E-16 | 6.59E+03 | 57.8 |
| **13** | **1.55E-15** | **1.57E+04** | **31.8** | 44 | 3.82E-15 | 3.82E+04 | 31.8 |
| **14** | **6.56E-16** | **6.63E+03** | **50.1** | 45 | 4.21E-15 | 4.22E+04 | 31.8 |
| 15 | 3.86E-16 | 3.86E+03 | 70.8 | 46 | 4.57E-15 | 4.58E+04 | 19.5 |
| 16 | 7.32E-16 | 7.34E+03 | 40.9 | 47 | 2.34E-15 | 2.34E+04 | 21.5 |
| 17 | 4.46E-16 | 4.47E+03 | 57.8 | 48 | 1.61E-15 | 1.61E+04 | 27.9 |
| 18 | 1.24E-15 | 1.25E+04 | 37.9 | 49 | 3.49E-15 | 3.50E+04 | 40.9 |
| 19 | 5.32E-16 | 5.33E+03 | 50.1 | 50 | 3.15E-15 | 3.15E+04 | 18.2 |
| 20 | 7.62E-16 | 7.64E+03 | 31.7 | 51 | 7.76E-16 | 7.78E+03 | 70.8 |
| 21 | 1.44E-15 | 1.45E+04 | 37.9 | 52 | 1.71E-15 | 1.71E+04 | 26.9 |
| 22 | 9.16E-16 | 9.18E+03 | 37.9 | 53 | 2.51E-15 | 2.52E+04 | 23.1 |
| 23 | 2.41E-16 | 2.42E+03 | 70.8 | 54 | 1.51E-15 | 1.51E+04 | 33.5 |
| 24 | 2.60E-16 | 2.60E+03 | 50.1 | 55 | 5.62E-16 | 5.64E+03 | 57.8 |
| 25 | 9.00E-17 | 9.02E+02 | 100.0 | 56 | 2.23E-15 | 2.23E+04 | 29.0 |
| 26 | 9.94E-16 | 9.97E+03 | 33.5 | 57 | 1.77E-15 | 1.77E+04 | 33.5 |
| 27 | 9.45E-16 | 9.47E+03 | 40.9 | 58 | 4.66E-16 | 4.67E+03 | 57.8 |
| 28 | 1.15E-16 | 1.15E+03 | 100.0 | 59 | 5.11E-15 | 5.12E+04 | 21.5 |
| 29 | 1.25E-15 | 1.26E+04 | 44.8 | **60** | **7.94E-16** | **8.14E+03** | **50.1** |
| 30 | 3.52E-16 | 3.53E+03 | 70.8 | **61** | **9.25E-16** | **9.48E+03** | **50.1** |
| 31 | 2.28E-15 | 2.28E+04 | 25.2 | | | | |

**In Bold**: These values represent the 8 blanks used for the single-batch blank correction and for the average blank correction.



**Table 2. List of the 57 sample values used to test our proposed approach. Note that the column marked as "Minimum $^{10}$Be content" indicates the value to use for the comparison with the LOD and LOQ values.**

| Sample Num | $^{10}$Be/$^9$Be | 1σ [%] | $^{10}$Be atoms uncorrected | 1σ atoms | Minimum $^{10}$Be content | Sample Num | $^{10}$Be/$^9$Be | 1σ [%] | $^{10}$Be atoms uncorrected | 1σ atoms | Minimum $^{10}$Be content |
|---|---|---|---|---|---|---|---|---|---|---|---|
| 1 | 1.60E-15 | 27.9 | 1.61E+04 | 4.50E+03 | 1.16E+04 | 31 | 4.66E-14 | 6.75 | 4.78E+05 | 3.23E+04 | 4.46E+05 |
| 2 | 1.99E-15 | 25.18 | 2.04E+04 | 5.50E+03 | 1.49E+04 | 32 | 4.95E-14 | 5.9 | 5.01E+05 | 2.94E+04 | 4.72E+05 |
| 3 | 2.02E-15 | 26.9 | 2.02E+04 | 5.09E+03 | 1.51E+04 | 33 | 4.97E-14 | 6.63 | 5.10E+05 | 3.38E+04 | 4.76E+05 |
| 4 | 3.44E-15 | 19.48 | 3.48E+04 | 6.79E+03 | 2.81E+04 | 34 | 7.41E-14 | 4.94 | 7.50E+05 | 3.70E+04 | 7.13E+05 |
| 5 | 3.75E-15 | 17.41 | 3.80E+04 | 6.62E+03 | 3.14E+04 | 35 | 8.34E-14 | 5.45 | 8.47E+05 | 4.61E+04 | 8.00E+05 |
| 6 | 4.14E-15 | 17.17 | 4.21E+04 | 7.23E+03 | 3.49E+04 | 36 | 1.23E-13 | 4.85 | 1.24E+06 | 6.04E+04 | 1.18E+06 |
| 7 | 4.73E-15 | 20.63 | 4.77E+04 | 9.84E+03 | 3.79E+04 | 37 | 1.63E-13 | 4.44 | 1.66E+06 | 7.37E+04 | 1.59E+06 |
| 8 | 4.75E-15 | 15.05 | 4.84E+04 | 8.31E+03 | 4.01E+04 | 38 | 2.24E-13 | 3.95 | 2.28E+06 | 9.00E+04 | 2.19E+06 |
| 9 | 4.79E-15 | 17.17 | 4.82E+04 | 7.25E+03 | 4.09E+04 | 39 | 2.80E-13 | 3.9 | 2.84E+06 | 1.11E+05 | 2.73E+06 |
| 10 | 4.92E-15 | 15.55 | 5.06E+04 | 8.81E+03 | 4.18E+04 | 40 | 2.81E-13 | 3.7 | 2.84E+06 | 1.05E+05 | 2.74E+06 |
| 11 | 5.01E-15 | 17.41 | 4.97E+04 | 7.73E+03 | 4.20E+04 | 41 | 3.17E-13 | 3.6 | 3.22E+06 | 1.16E+05 | 3.11E+06 |
| 12 | 5.20E-15 | 16.29 | 5.28E+04 | 8.61E+03 | 4.42E+04 | 42 | 4.03E-13 | 3.5 | 4.07E+06 | 1.42E+05 | 3.92E+06 |
| 13 | 6.23E-15 | 13.47 | 6.33E+04 | 8.53E+03 | 5.48E+04 | 43 | 4.15E-13 | 3.79 | 4.21E+06 | 1.60E+05 | 4.05E+06 |
| 14 | 6.72E-15 | 17.4 | 6.80E+04 | 1.18E+04 | 5.61E+04 | 44 | 4.27E-13 | 3.5 | 4.31E+06 | 1.52E+05 | 4.16E+06 |
| 15 | 7.53E-15 | 14.46 | 7.60E+04 | 1.10E+04 | 6.50E+04 | 45 | 4.28E-13 | 3.5 | 4.32E+06 | 1.53E+05 | 4.17E+06 |
| 16 | 8.37E-15 | 15.38 | 8.60E+04 | 1.32E+04 | 7.27E+04 | 46 | 5.39E-13 | 3.49 | 5.53E+06 | 1.93E+05 | 5.33E+06 |
| 17 | 8.86E-15 | 13.2 | 8.94E+04 | 1.18E+04 | 7.76E+04 | 47 | 5.78E-13 | 3.35 | 5.85E+06 | 1.96E+05 | 5.66E+06 |
| 18 | 6.21E-15 | 12.9 | 9.38E+04 | 1.21E+04 | 8.17E+04 | 48 | 5.92E-13 | 3.4 | 5.98E+06 | 2.04E+05 | 5.78E+06 |
| 19 | 1.01E-14 | 14.1 | 1.02E+05 | 1.43E+04 | 8.73E+04 | 49 | 5.95E-13 | 3.32 | 6.03E+06 | 2.00E+05 | 5.83E+06 |
| 20 | 1.01E-14 | 12.67 | 1.02E+05 | 1.30E+04 | 8.94E+04 | 50 | 8.88E-13 | 3.29 | 9.04E+06 | 2.97E+05 | 8.74E+06 |
| 21 | 1.12E-14 | 13.3 | 1.13E+05 | 1.49E+04 | 9.77E+04 | 51 | 9.03E-13 | 3.3 | 9.16E+06 | 3.02E+05 | 8.86E+06 |
| 22 | 1.31E-14 | 11.51 | 1.32E+05 | 1.52E+04 | 1.17E+05 | 52 | 9.30E-13 | 3.3 | 9.40E+06 | 3.06E+05 | 9.09E+06 |
| 23 | 1.61E-14 | 16.29 | 1.63E+05 | 2.66E+04 | 1.37E+05 | 53 | 1.18E-12 | 3.25 | 1.19E+07 | 3.88E+05 | 1.15E+07 |
| 24 | 1.68E-14 | 10.1 | 1.69E+05 | 1.71E+04 | 1.52E+05 | 54 | 2.23E-12 | 3.1 | 2.25E+07 | 6.99E+05 | 2.18E+07 |
| 25 | 1.78E-14 | 10.69 | 1.80E+05 | 1.92E+04 | 1.61E+05 | 55 | 2.38E-12 | 3.12 | 2.42E+07 | 7.56E+05 | 2.35E+07 |
| 26 | 1.83E-14 | 10.2 | 1.84E+05 | 1.88E+04 | 1.66E+05 | 56 | 2.94E-12 | 3.11 | 2.98E+07 | 9.26E+05 | 2.89E+07 |
| 27 | 1.87E-14 | 10.0 | 1.89E+05 | 1.90E+04 | 1.70E+05 | 57 | 4.20E-12 | 3.08 | 4.27E+07 | 1.31E+06 | 4.13E+07 |
| 28 | 2.15E-14 | 10.55 | 2.21E+05 | 2.33E+04 | 1.98E+05 | | | | | | |
| 29 | 2.03E-14 | 7.82 | 2.45E+05 | 1.92E+04 | 2.26E+05 | | | | | | |
| 30 | 4.45E-14 | 7.2 | 4.58E+05 | 3.29E+04 | 4.25E+05 | | | | | | |



**Table 3. Limit of Detection (LOD) and Limit of Quantification (LOQ) for the eight blank-ensemble (processed along with the samples) and the 61 GFZ long-term blanks. The subscript "N" after LOD and LOQ refers to the normal distribution whereas the "NB" refers to the negative-binomial distribution.**

| | $LOD_N$ Atoms (x10$^4$) | $LOQ_N$ atoms (x10$^4$) | $LOD_{NB}$ atoms (x10$^4$) | $LOQ_{NB}$ atoms (x10$^4$) |
|---|---|---|---|---|
| Eight blanks (n=8) | 1.93 | 4.65 | 2.39 | 4.78 |
| Long-term blanks (n=61) | 7.59 | 21.31 | 11.45 | 30.20 |
| Confidence interval (%) | 99.9 | 100 | 99.9 | 100 |
| **Number of samples falling below the given limit** | | | | |
| Eight blanks (n=8) | 3 | 12 | 3 | 12 |
| Value in % | 5.3 | 21.1 | 5.3 | 21.1 |
| Long-term blanks (n=61) | 16 | 28 | 21 | 29 |
| Value in % | 28.1 | 49.1 | 36.8 | 50.9 |




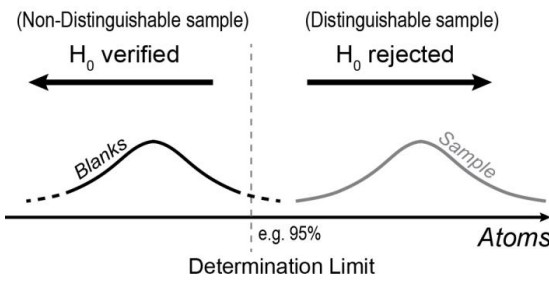

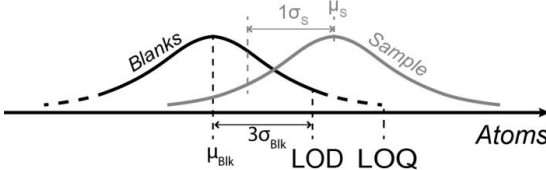

**Figure 1. Examples of samples with high (a) and low (b) $^{10}$Be content. The black curves describe the distribution of several individual blanks, while the grey curves represent one single sample. As an example, we choose a confidence interval of 95% for setting the determination limit. Samples whose lower uncertainty value (i.e., $\mu_S - \sigma_S$) is below the**
5     **determination limit are consistent with the null hypothesis ($H_0$), and as such they cannot be distinguished from the blanks; in contrast, values above the limit will reject the null hypothesis, thus being statistically distinguishable from the blanks. a) For high $^{10}$Be content, the samples' values are more likely to lie above the limit, resulting in a relatively low probability of accepting $H_0$. b) For a sample with low $^{10}$Be content, the probability of falling below the determination limit is larger and it increases when increasing the chosen confidence interval for the blank distribution (i.e., choosing**
10     **the LOQ rather than the LOD). In the example proposed, the lower limit of the sample's value ($\mu_S - 1\sigma_S$) falls below both the LOD and LOQ limits.**





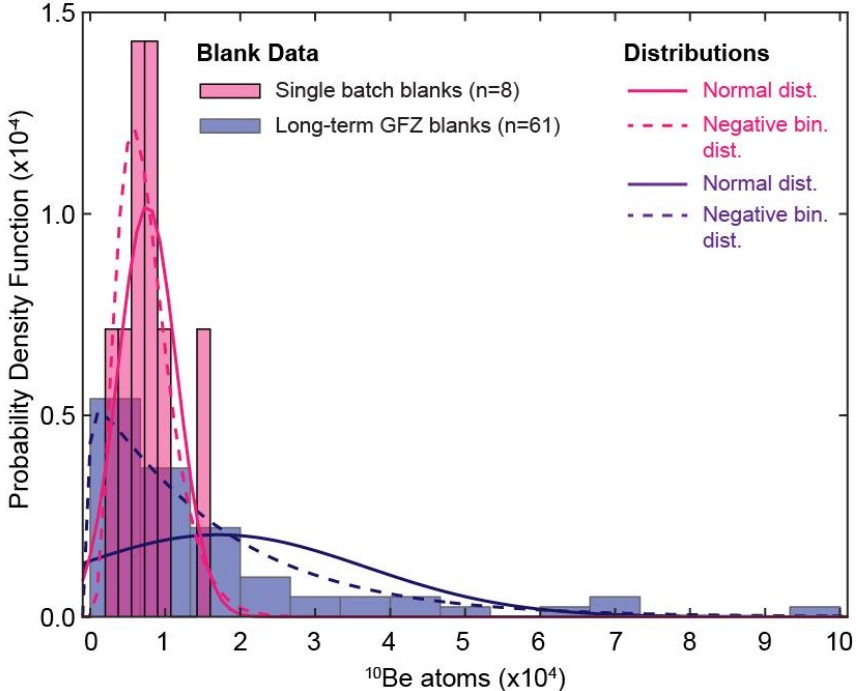

**Figure 2.** Probability density function (PDF) and normalized frequency of the eight single blanks (in pink, histogram) approximated with a normal distribution (in pink, solid line) and with a negative binomial distribution (in pink, dashed line). PDF and normalized frequency of the 61 long-term blanks from the GFZ (in violet, histogram) represented by both a normal distribution (violet, solid line) and a negative binomial distribution (violet, dashed line). Probability distribution parameters associated with the normal distribution are (in $^{10}$Be atoms): $\mu = (0.76 \pm 0.14) \times 10^4$ for the single batch blank method; $\mu = (1.72 \pm 0.25) \times 10^4$ for the long-term GFZ blanks. When using the Negative-Binomial distribution, the rate of success is $R = 4.37 \pm 2.11$ and the probability is $P = (57 \pm 29) \times 10^{-5}$ for the eight blanks, and $R = 1.06 \pm 0.17$ and the probability is $P = (6 \pm 1) \times 10^{-5}$ for the long-term GFZ values. The best fit test with the Akaike Information Criterion (AIC) (Cavanaugh and Neath, 2011) indicates that both the blank datasets are better described by a negative binomial distribution (smaller AIC). For the 8 blanks: AIC (normal distribution) = 161.19, AIC (negative binomial distribution) = 159.64 for the negative binomial. For the 61 long-term blanks: AIC (normal distribution) = $1.38 \times 10^3$, AIC (negative binomial distribution) = $1.31 \times 10^3$.





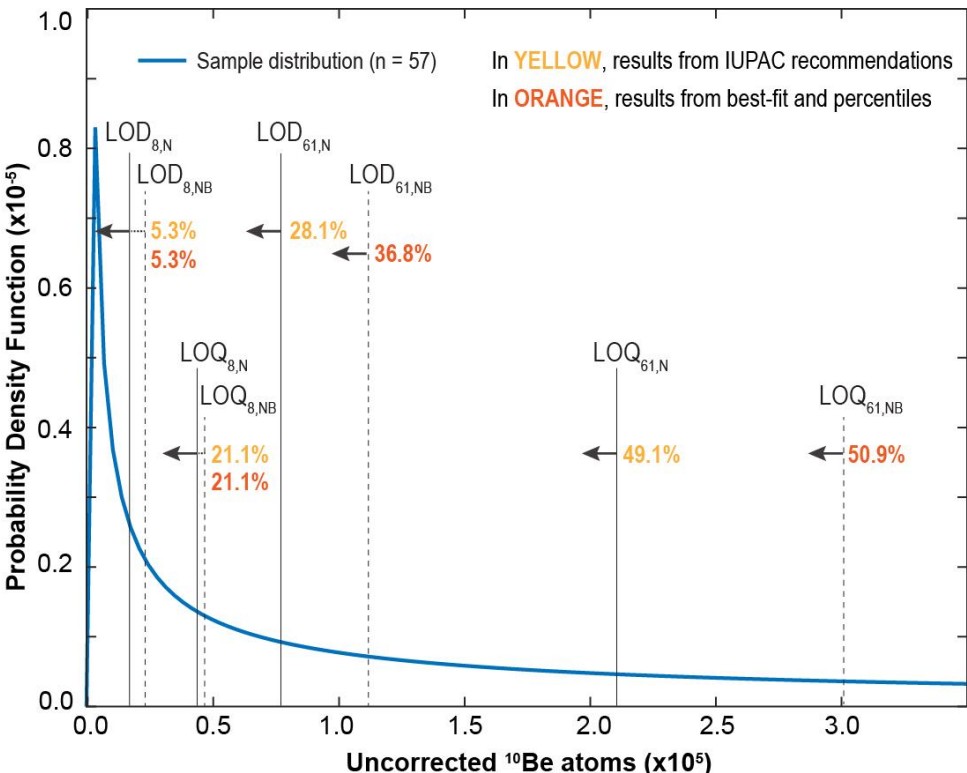

**Figure 3. Probability Density Function (PDF) of the 57 samples with calculated determination limits. Note that the samples are best represented by a negative binomial distribution (blue curve; displaying only values < 3.5 x 10$^5$ atoms for better visibility of the limits). Numbers at the side of the arrows represent the percentage of samples falling below the assigned limit. LOD and LOQ are calculated using Eq. (2) and (3) when using the IUPAC's recommendation (subscript "N" for normal distribution), and, equivalently, at the correspondent percentiles when using the best-fit negative binomial distribution (subscript "NB", see Table 3 for relative values).**




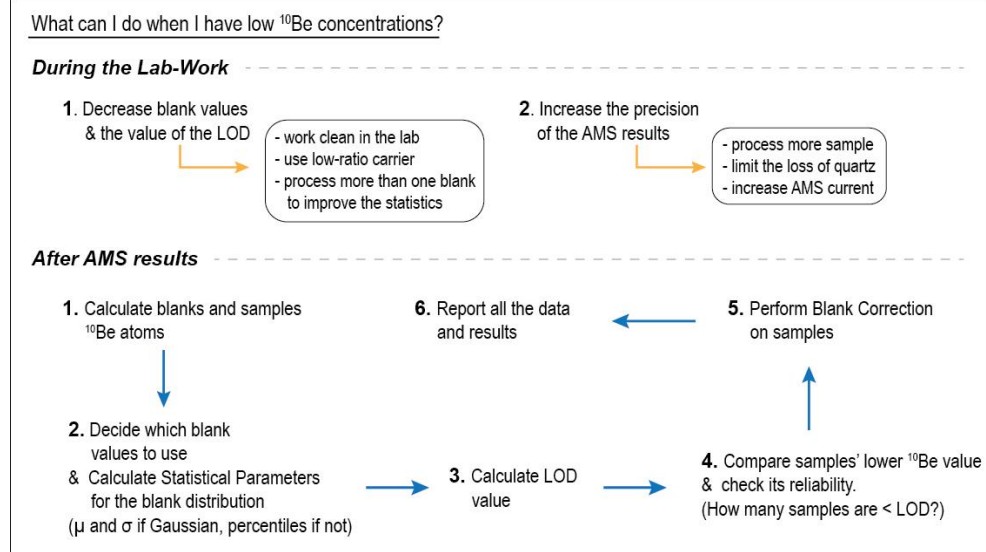

**Figure 4. Schematic view of the steps to perform in case of samples with low ¹⁰Be content. Decreasing the blank values and increasing the precision of the AMS measurements allow to have smaller thresholds and more precise data with small uncertainties.**




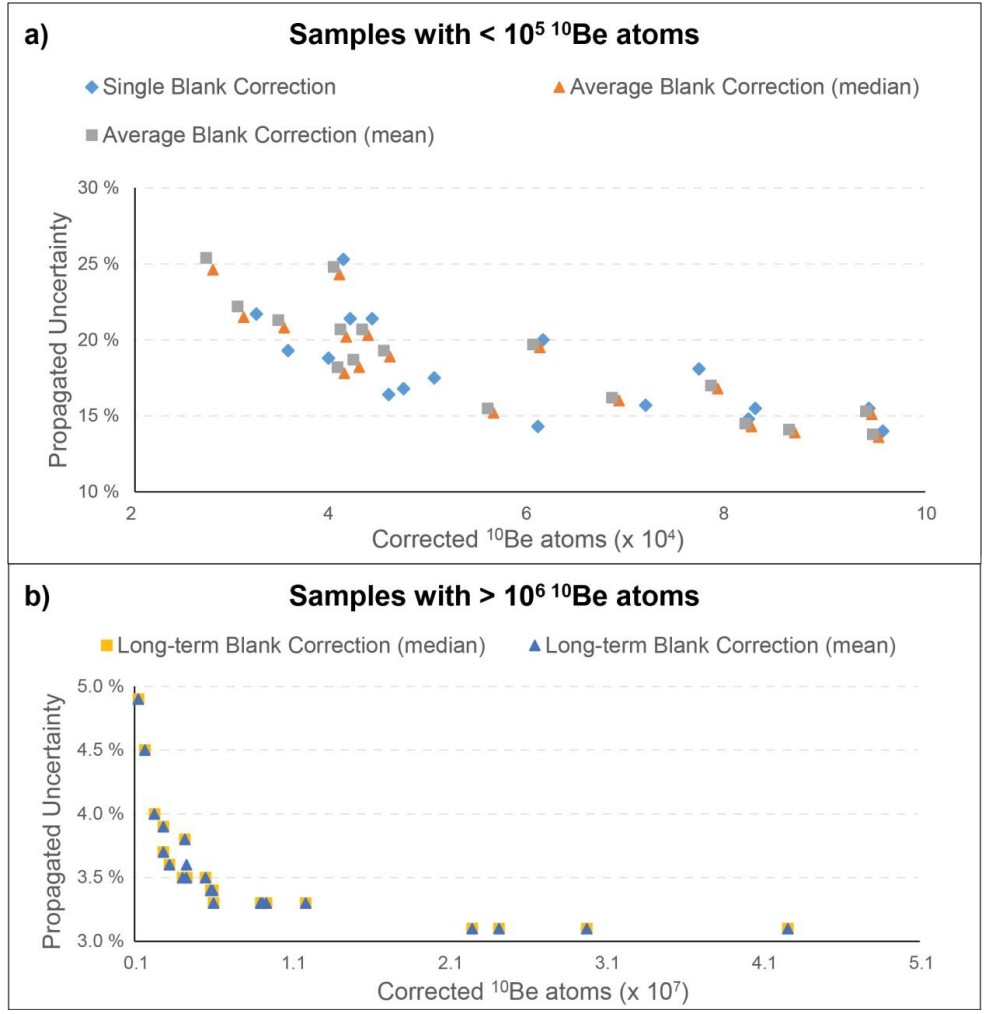

**Figure 5. This figure illustrates the difference in values and related uncertainties of the corrected samples associated with the different blank correction methods. a) Samples with low $^{10}$Be atoms (i.e., $< 10^5$) are more sensitive to the applied correction method (note that all samples corrected with the long-term blank correction are not distinguishable from the blanks for values $< 10^5$ $^{10}$Be atoms). b) Samples with high $^{10}$Be atoms (i.e., $> 10^6$) do not show significant differences (note that the samples corrected with the single blank correction and average blank correction methods are not visible because their values overlap with the values obtained from the long-term blank correction).**