# Peer review of "Determination limits for cosmogenic 10Be and their importance for geomorphic applications"

_Earth Surface Dynamics, 2017_

## Referee Comment (RC1) · Anonymous Referee #1 · 20 Jun 2017

Summary: This ms applies what are considered standard approaches in analytical chemistry to determining detection limits for the analysis of cosmogenic 10Be. The ms uses a long term set of blanks and sample measurements from one lab but multiple operators to make calculations and determine several statistical parameters for the detection of 10Be at levels confidently above the blank.

The manuscript does not break new ground; it is an exercise done to understand the potential limits of one laboratory much more than a significant advance in cosmogenic nuclide science and therefore, not the type or style of paper for publication in a broad readership journal such as Esurf. Overall, this paper presents a data set that most cosmogenic labs have and have likely analyzed internally but the ms does little to advance cosmogenic nuclide science more broadly. As a referee and as a user of this

literature, I consider the ms to be much more appropriate for an AMS-specific journal such as NIMs and as such suggest it be shortened and submitted for publication there rather than eSurf. It just does not fit well nor will be it be of interest to most of the surface process community. It is technical in nature and does not have any significant geomorphic impact.

Furthermore, the ms is proscriptive and narrow in its approach, which makes it less likely to be accepted by the community. The manuscript does not set the presented data in context because it does not include a critical evaluation of how previous workers have done blank corrections nor does it demonstrate how different blank corrections would change geomorphic outcomes of extant studies. There are several examples in the recent literature where workers have applied several different approaches to blank correction and tested the sensitivity of results to varying approaches; see for example Corbett et al., GRL (2017) on 10/26 ratios. These are not cited nor are they considered critically. The ms reviewed here is under-referenced, omitting numerous important citations both classical and recent that are germane.

Critically, the paper does not consider type 1 vs type 2 errors, that is in striving to be certain that 10Be is confidently detected, such as using 99.9% confidence, samples containing real 10Be are almost certainly being rejected. This will lead to errant science and must be considered head on in any revision before publication. It is a critical flaw and must be addressed before publication in any journal.

Lastly, boron, an isobaric interference is neglected. It varies sample to sample and the means by which it is rejected varies between AMS facilities. It can be a very real component of the blank measurements. At minimum, it needs to be presented and discussed in the context of the ms and the measurements within – better yet would be to consider it broadly across the community. Again, this is narrow, technical information of interest to a small section of the community but critical to the issues here.

In summary, the ms is not appropriate for Esurf, is too narrowly focused on one approach to blank subtraction, ignores Type 1 vs Type 2 errors, does not consider 10B interference, and has little critical evaluation of the literature in which the current data need be considered in context. As is, it is a formulaic approach to analyzing an isotopic rather than geomorphic data set and not a significant advance of geomorphic science.

Suggestions for improvement:

I would encourage authors to expand their selection of references in particular citing more studies that make very low level measurements and how these studies have dealt with blanks as well to cite some of the many excellent review papers since 2010 that compile both erosion rate and exposure age data. For example, Carlson and Nelson have recently both published very low concentration measurements from Greenland, one for glacial dating and the other for sediment tracing and neither of these studies are cited.

P2, Ln 25: most AMS measurements in the 10-15 and 10-16 range are not very precise, reword.

P2, Ln 27: This would be an appropriate place to cite Corbett et al. (2016) who review in great detail lab and AMS issues affecting detection including blanks and AMS beam currents. They in turn cites others such as the work on precision by Rood at the LLNL facility.

P2, Ln 30-35: It is not clear why a standardized approach is needed or an improvement on the current approach; there is so much buried in blanks including which AMS, operator changes, real contamination. The paper would likely be better accepted by the community if it were to provide a means or variety of means by which the blank subtraction could be done. The way this section is written presumes the authors have defined "the" way to do blank correction not "a" way to do blank correction. This will not advance the science of AMS and 10Be.

P3, Ln 10: This sentence is incorrect. 10Be IS naturally present in earth materials.

P3, Ln 17: This set of references fails to cite the original 3 references for nuclides in sediments – a critical oversight that needs to be remedied.

P4, Ln1: This sentence omits an important part of running blanks, determining the dark current or background of the AMS system including cross talk in the source. This needs to be mentioned and cited properly.

P4, Ln 5-8: This is at least not correct for our lab and I think not correct for other labs. For us, every sample including blanks gets the same reagent amount and same open beaker time. Otherwise, how could we could compare process blanks and samples and do the subtraction in a meaningful way?

P4, ln 15-20: This section omits a critical issue in defining blanks on the AMS – boron as an isobaric interference and how that is handled. The process varies between AMS (some such as PRIME use GFM that completely removes 10B, others like LLNL make a very uncertain correction, others use post stripping). For low level samples, this is critical.

P4, ln 21-22: This is one way to do it but if the same amount of carrier is added to all samples, then the ratios can be subtracted. This alternative approach needs to be acknowledged and cited.

P4, ln 36: there needs to be more here. How do blanks change over time? Does contamination increase over time in labware?

P5, Ln 7-10: This is far too simplistic. Only considering the upper value of the blank will result is rejecting data that are likely real. The blank subtraction process is a probabilistic one and different for different purposes. The goal of determining whether something is confidently detected is very different than the goal of best estimating the blank for subtraction. The paper would be much stronger if it considered this subtlety.

P6, Ln 4: Unmentioned here is the fact that blanks are very imprecise measurements by their nature and because of the Poisson distributed counting statistics of AMS. The
lowest blanks can contain only a count or two (or even none). Implicit in the discussion above is that the blanks are normally distributed (parametric statistics). If these issues were discussed, the paper would be much stronger. Some of this discussion is in section 3 and could be moved up.

P8, Ln 5: "Although a minimum of 20 values" this has been stated repeatedly. No need to state again.

P8, Ln 35: It is not clear how the average blank constrains any temporal variance? Please explain.

P9, ln 18: I find this paragraph very hard to follow and not very informative. A table or graphic would convey the same information much more effectively. Some of the information is summarized in tables but not all of it.

P11, ln 13: This is a critical mis-understanding of Type 1 and Type 2 errors. Here, the authors suggest that, "In general, the use of the long-term laboratory blanks (being based on many blank measurements) guarantees more reliable values for the statistics of the blank distribution and for the calculation of the determination limits; as such, they may be preferred." The approach the authors suggest is very likely to introduce errors of rejection for data (samples) that contain actual 10Be above blank levels.

Section 5.4: This is not an adequate critical review of what others have done with low activity samples.

The data tables for AMS would be much more informative if they included the standard(s) to which the ratios were normalized, more about the boron counts and rejection procedures, comparison of sample beam currents to standard beam currents, the number of gated 10Be counts, and the actual uncertainties of the measured ratios.

---

## Referee Comment (RC2) · Anonymous Referee #2 · 21 Jun 2017

Earth Surface Dynamics Discussion: Determination limits for cosmogenic 10Be and their importance for geomorphic applications

In this manuscript, Savi and colleagues review and compare the procedures that are typically used to correct AMS 9Be/10Be measurement for laboratory blanks contributions. The paper discusses the effects of considering i) long-term, inter-operator lab blanks, ii) long-term single-operator lab blanks and iii) the blanks that are processed during a single sample batch on low 10Be concentration samples. This discussion is based on a large blank data-set produced by the GFZ Potsdam group over the past years.

The paper is very clear, well written and the number of blanks upon which the discussion is based is significant enough to support the discussion. The statistical approach,

albeit simple, seems robust and well discussed. Overall the merits of this paper lays in the fact that it is the first paper to my knowledge that specifically discusses blank corrections in the 10Be community. That said, I have some concerns with respect to the publication of this paper in ESurf: eventhough it is a useful contribution, I believe that all procedures described here are pretty standard for any analytical measurement made in the geosciences and one would hope that such a reasoned blank correction approach is already widely applied across 10Be labs. I therefore question the novelty and impact of this contribution for a broader community outside of the cosmogenic nuclide people, who should in principle already be aware of these issues in the case of low 10Be concentration samples. In my opinion the paper could be improved and made more significant by having a more systematic evaluation of the sources of blank contamination and the methods to deal with it throughout the entire Be preparation and measurement procedure, from lab to the AMS measurement.

A few things that could be included or need to be better discussed:

- First, I am also a bit puzzled by the different levels of confidence that you are mixing in the manuscript: you consider the limit of detection basically at a 3-sigma level (99.7%) but then compare that to a sample 10Be/9Be measurement that is given at a 1-sigma level (68.3 %). Wouldn't it be preferable to be consistent for all types of measurements you are considering?

- I would like to see the equations that are used to do the actual blank correction (do you directly consider 9/10 ratios or the number of 10Be atoms as this may yield slightly different results if the amount of carrier that was used is not constant across all measurements).

- There is no discussion about the low uncertainty that are associated with blank measurements and how that affects the correction. Also, how 10B isobaric interferences are corrected for in the case of low count blank determinations is something that should be mentioned and discussed.

**ESurfD**
- A broader discussion that includes the whole preparation and measurement procedure would for instance investigate: o the impact of reducing carrier to quartz ratios in terms of overall uncertainty and blank assessment as this increases 10/9 ratios but decreases measurement time or 9Be currents... o the influence of isobaric interferences (10B) on the low ratio measurements... o the relative contribution of the blank correction method to the overall uncertainty and reproducibility of low 10Be concentration measurements (for instance comparing it to CRONUS or internal standard measurements) o What is the impact of very low 10/9 Be ratio carriers on the final measurement (i.e. if the variability is still as high as for commercial, higher ratio carriers).

I hope this helps to further improve the manuscript.

**ESurfD**

---

## Author Comment (AC1) · 29 Jun 2017

*Answer to Anonymous Referee #1, summary comments*: Thanks for the detailed review. We understand the perspective of Referee #1 with respect to the technicality of the paper and its broader interest for the surface processes community. However we don't agree completely with his/her assertions.

We would like to highlight that in this paper we describe a statistical method that defines a lower threshold useful for the interpretation of low-concentration samples. This method is known in analytical chemistry but has never been applied to cosmogenic studies. The method is designed to be used by the final user, after the chemical procedure and the AMS measurements of the samples, and it is not simply an "*exercise done to understand the potential limits of one laboratory*". Rather it is designed to help the cosmogenic nuclide community users defining the geomorphic meaning of samples with low nuclide concentration. In no way we suggest that the results which do not lie significantly above the blank threshold cannot be used, but rather we use this threshold to determine whether or not a rate or age can be quantified or when we can only interpret the data in terms of a maximum age or minimum erosion rate.

In the following lines we will proceed with commenting the main concerns of Referee #1 (in black ink), especially considering the aim of our work and our targeted audience, which define the structure and meaning of our paper. A more detailed answer on the line-by-line comments and changes to the manuscript will follow in the next weeks.

**Anonymous Referee #1**: Summary: This ms applies what are considered standard approaches in analytical chemistry to determining detection limits for the analysis of cosmogenic 10Be. The ms uses a long term set of blanks and sample measurements from one lab but multiple operators to make calculations and determine several statistical parameters for the detection of 10Be at levels confidently above the blank.

The manuscript does not break new ground; it is an exercise done to understand the potential limits of one laboratory much more than a significant advance in cosmogenic nuclide science and therefore, not the type or style of paper for publication in a broad readership journal such as Esurf. Overall, this paper presents a data set that most cosmogenic labs have and have likely analyzed internally but the ms does little to advance cosmogenic nuclide science more broadly. As a referee

and as a user of this literature, I consider the ms to be much more appropriate for an AMS-specific journal such as NIMs and as such suggest it be shortened and submitted for publication there rather than eSurf. It just does not fit well nor will be it be of interest to most of the surface process community. It is technical in nature and does not have any significant geomorphic impact.

*Significance and Readership*: First of all, we would like to highlight that the aim of this paper is to highlight the issues that one encounters with low concentration samples (i.e., $< 10^5$) and to provide a standardized method for the interpretation of the results. The paper is designed for the cosmogenic nuclide final users, including all the people that use the cosmogenic nuclide technique for studying earth-surface processes, but who are not necessarily familiar with the technicality of the Be-extraction/measurement procedure. In the last decades, the rapid expansion in the use of this technique has created a growing community of users that apply cosmogenic studies without the detailed technical knowledge typical of the people that work in an institute where it is possible to perform Be-extraction and/or AMS measurements (e.g., the LLNL facility) and who may have only a basic knowledge of the technical aspects of these procedures. This paper aims also at this broader community and, as such, it is specifically designed not to be technical, since we want it to be suitable for non-technical users. Some technical information on the chemical procedure for quartz extraction, Be-isolation, and the possible sources of blank contamination are, however, additionally provided in the supplementary material. Other detailed information on very specific technical aspects, such Boron interference, background AMS beam current, and blank counts, are already central topics of existing and excellent technical papers that the readers can access separately, if interested (we provide some recent summary papers as examples, e.g., Balco, 2011; Granger et al., 2013; and some technical papers that can be of interest for the more curious users, e.g., Schaefer et al., 2009; Rood et al., 2010; Merchel et al., 2013; Portenga et al., 2015; Corbett et al., 2016). Most of this technical information, indeed, is available for internal usage of specific laboratories (e.g., to define the precision of the AMS machine and/or cleanliness of the laboratory) but are normally not included in the final AMS measurement report that is delivered to the users.

For the above mentioned reasons, our paper does not deal with technical aspects of the AMS measurements (for which, we believe, already exists more suitable end technical papers), but rather proposes a statistical procedure (not chemical nor isotopic) based on clear definitions of detection limits which can be used as standard reference in the interpretation of the cosmogenic

measurements. As such, the paper targets a broad audience and it describes a method that can be applied by all the cosmogenic nuclide users. Specifically, the method provides a statistical threshold to evaluate whether the nuclide concentration can be used to quantify an exposure age or erosion rate versus only limit the age or rate; in the latter case, the data is not rejected (nor there is the risk to *reject data that are likely real*); on the contrary, a limiting value can still be very helpful. This proposed method, additionally, can be used also in cases where low-concentration samples are not expected. In such cases the user may not be aware of the precautions necessary to be taken during the chemical procedure, but he/she will still need to define how the results should be interpreted. For this goal, the paper describes a procedure that is commonly used in analytical chemistry to define a statistically significant threshold for low-concentration samples and provides a method for its application in the field of cosmogenic nuclide studies. Consequently, it does not describe a "new procedure", but rather we propose to apply this standardized method for cosmogenic nuclide studies, seeing as such a procedure is so far missing in the community. This aspect is very relevant, especially considering that always more researchers will have to deal with low concentration samples (see Corbett et al., 2016). We believe that a common and standardized approach that defines a general rule for the interpretation of these data is timely and valuable and that can be of interest for a broad audience and not only for the readers of a technical journal such as NIMs. For these reasons, we strongly believe that the paper will fit well a journal such as ESurf.

Furthermore, the ms is proscriptive and narrow in its approach, which makes it less likely to be accepted by the community. The manuscript does not set the presented data in context because it does not include a critical evaluation of how previous workers have done blank corrections nor does it demonstrate how different blank corrections would change geomorphic outcomes of extant studies. There are several examples in the recent literature where workers have applied several different approaches to blank correction and tested the sensitivity of results to varying approaches; see for example Corbett et al., GRL (2017) on 10/26 ratios. These are not cited nor are they considered critically. The ms reviewed here is under-referenced, omitting numerous important citations both classical and recent that are germane.

With respect to the blank subtraction methods, we have presented here three different approaches and discussed how their application would change the final interpretation of the results. So we do

not agree with Referee #1 when he/she said that we did not consider different blank subtraction methods or that we did not "*demonstrate how different blank corrections would change geomorphic outcomes of extant studies*". Nevertheless, in the revised version of the manuscript, we will add more information on how other people have dealt with this topic (e.g., Corbett et al., as suggested by Referee #1) and discuss their approach. We would also like to highlight that very few papers discussed and reported in detail how the blank correction has been performed, and even fewer papers additionally dealt with low concentration samples (i.e., $< 10^5$ $^{10}$Be atoms). As such, this information is not so easy to find within the existing literature.

This point raised from Referee #1 demonstrates how a common procedure for these important steps in the calculation of the cosmogenic results is still missing within the community. The way people normally deal with blank corrections is often not reported in the papers, but this could lead to slightly different results and to a lack of comparability between different publications. This highlights once more why our paper will represent an important contribution for the cosmogenic nuclide community.

Also, we have already provided, as examples, some of the most recent summary papers (e.g., Balco, 2011; Granger et al., 2013) that provide an exhaustive explanation of the cosmogenic nuclide technique and its last advances. The interested readers can go into the details of the references mentioned within these papers, based on their own specific interests. In the revised version of the manuscript, we will had more references to not leave out some of the very recent papers (e.g., Corbett et al., 2017), as suggested by Referee #1. However, we would also like to point out that we have provided here the reference to some of the most important and classical papers, but that it is possible that we have missed some of the very recent works (e.g., Corbett et al., 2017) that have been published only few months before the submission of this paper.

Critically, the paper does not consider type 1 vs type 2 errors, that is in striving to be certain that 10Be is confidently detected, such as using 99.9% confidence, samples containing real 10Be are almost certainly being rejected. This will lead to errant science and must be considered head on in any revision before publication. It is a critical flaw and must be addressed before publication in any journal.

*As already mentioned above, for the aim of our work and the targeted readership, the paper is specifically designed not to be overly technical. For the same reasons, we did not go into the details of the Type 1 and Type 2 statistical errors: these are basic statistical notions that can be found in any book of statistics and we do not believe that a discussion about these types of errors will bring any benefit for the readers nor it will change the procedure/discussion presented in the paper. Additionally, we highlight once more that in no way "samples containing real 10Be are going to be rejected", rather we suggest that they can be interpreted as limiting values for exposure ages or erosion rates, a still very helpful and meaningful result.*

Lastly, boron, an isobaric interference is neglected. It varies sample to sample and the means by which it is rejected varies between AMS facilities. It can be a very real component of the blank measurements. At minimum, it needs to be presented and discussed in the context of the ms and the measurements within – better yet would be to consider it broadly across the community. Again, this is narrow, technical information of interest to a small section of the community but critical to the issues here.

*We agree that this is a "narrow, technical information of interest to a small section of the community" and for the reasons already mentioned above, we did not go into the detail of this discussion. Additionally, as pointed out by the Referee, this is an aspect that changes from AMS to AMS laboratories and its discussion goes far outside the aim of our paper and our targeted audience. Nevertheless, we have reported a full paragraph in the supplementary material where we have discussed the issue of the Be contamination (Section S5). We will had there some more information about Boron interference giving some more reference as examples for the interested users.*

In summary, the ms is not appropriate for Esurf, is too narrowly focused on one approach to blank subtraction, ignores Type 1 vs Type 2 errors, does not consider 10B interference, and has little critical evaluation of the literature in which the current data need be considered in context. As is, it is a formulaic approach to analyzing an isotopic rather than geomorphic data set and not a significant advance of geomorphic science.

We have presented here three types of blank subtraction, an aspect that is not normally discussed in cosmogenic papers. We will provide additional information using the work of Corbett et al. to include other possible approaches. We have explained why our paper does not want to be too technical (still, many technical information can be found in the supplementary material), considering the aim and the target audience for this work, and we have provided two examples of how our proposed approach can be applied to geomorphic studies, discussing its consequences for the geomorphic interpretation of the samples (section 5.4). In the revised version of the paper we will include more examples, as suggested by Referee #1, so that the readers can have a broader idea of the impact of the proposed method on geomorphic studies.

For all these reasons, we believe that the paper is of broad interest and that it presents timely and necessary information for the cosmogenic nuclide community and for a correct interpretation of low-concentration samples. Its publication in a broad journal such as ESurf, rather than in a technical journal, will surely be of benefit for many researchers within and outside the cosmogenic nuclide community.

---

## Author Comment (AC2) · 29 Jun 2017

Answer to Anonymous Referee #2, summary comments: Thanks for the detailed review. We would like to highlight that in this paper we do not only "review and compare the procedures that are typically used to correct AMS 9Be/10Be measurement for laboratory blanks contributions". This is, indeed, only one part of the paper, but the real focus of our work is the description of a statistical method that defines a lower threshold useful for the interpretation of low-concentration samples. This method is known in analytical chemistry but has never been applied to cosmogenic studies, and the contribution of our work to the community is to present how to apply this standardized method and use it to interpret cosmogenic data. The method is designed to be used by the final user and wants to help the cosmogenic nuclide community users defining the geomorphic meaning of samples with low nuclide concentration. Our comments follow the Referee's comments, which are left in black ink.

**Anonymous Referee #2**: In this manuscript, Savi and colleagues review and compare the procedures that are typically used to correct AMS 9Be/10Be measurement for laboratory blanks contributions. The paper discusses the effects of considering i) long-term, inter-operator lab blanks, ii) long-term single-operator lab blanks and iii) the blanks that are processed during a single sample batch on low 10Be concentration samples. This discussion is based on a large blank dataset produced by the GFZ Potsdam group over the past years.

The paper is very clear, well written and the number of blanks upon which the discussion is based is significant enough to support the discussion. The statistical approach, albeit simple, seems robust and well discussed. Overall the merits of this paper lays in the fact that it is the first paper to my knowledge that specifically discusses blank corrections in the 10Be community. That said, I have some concerns with respect to the publication of this paper in ESurf: even though it is a useful contribution, I believe that all procedures described here are pretty standard for any analytical measurement made in the geosciences and one would hope that such a reasoned blank correction approach is already widely applied across 10Be labs. I therefore question the novelty and impact of this contribution for a broader community outside of the cosmogenic nuclide people, who should in principle already be aware of these issues in the case of low 10Be concentration samples. In my opinion the paper could be improved and made more significant by having a more systematic evaluation of the sources of blank contamination and the methods to deal with it

throughout the entire Be preparation and measurement procedure, from lab to the AMS measurement.

As already mentioned above and in the Answer to the Referee #1, the aim of this paper is to highlight the issues that one encounters with low concentration samples (i.e.,  $< 10^5$ ) and to provide a standardized method for the interpretation of the results. The paper is designed for the cosmogenic nuclide final users, including all the people that use the cosmogenic nuclide technique for studying earth-surface processes, but who are not necessarily familiar with the technicality of the Beextraction/measurement procedure. We agree with the Referee that "all (chemical) procedures described here are pretty standard for any analytical measurement made in the geosciences and one would hope that such a reasoned blank correction approach is already widely applied across 10Be labs". However, as pointed out by the Referee, this approach is not normally reported, or explained in detail, within standard cosmogenic papers (i.e., non-technical papers). This is one reason for which we report such a detailed description, describing three different possible methods for blank corrections and describing their implication for the results. However, this does not represent the "novelty and impact of this contribution", rather, the contribution made by this paper lies in the description of a procedure that is commonly used in analytical chemistry to define a statistically significant threshold for low-concentration samples. In this paper we propose to apply this standardized method for cosmogenic nuclide studies, seeing that such a procedure is so far missing in this community. This aspect is very relevant, especially considering that always more researchers will have to deal with low concentration samples (see Corbett et al., 2016) and such a standardized procedure could help having a common reference for the interpretation of the results. For these reasons, we believe that this paper is timely and valuable in its content and that can be of interest for a broader audience, also within the cosmogenic nuclide community. Indeed, the paper focuses on a statistical procedure that can be used by the final user who may only have basic technical information about the Be-extraction/measurement procedure, but who still has to evaluate the geomorphic meaning of the AMS measured samples. Specifically, the method provides a threshold to evaluate whether the nuclide concentration can be used to quantify an exposure age or erosion rate versus only limit the age or rate; in the latter case, the data is not rejected; on the contrary, a limiting value can still be very helpful and meaningful.

Also, an "evaluation of the sources of blank contamination and the methods to deal with it throughout the entire Be preparation and measurement procedure, from lab to the AMS measurement" is already provided in the supplementary material document associated to the main manuscript.

A few things that could be included or need to be better discussed:

- First, I am also a bit puzzled by the different levels of confidence that you are mixing in the manuscript: you consider the limit of detection basically at a 3-sigma level (99.7%) but then compare that to a sample 10Be/9Be measurement that is given at a 1-sigma level (68.3 %). Wouldn't it be preferable to be consistent for all types of measurements you are considering?

Thanks for this comments. This choice has been made with a specific reasoning: 1) The analytical procedure used in analytical chemistry mostly discusses the LOD and LOQ as corresponding to 3- and 10-sigma. These are the confidence levels normally attributed to these detection limits. To be consistent in the definition of "detection limits", we describe the procedure with the same confidence levels. However, we also wrote in the paper that "when such high confidence is not required, the users may alternatively use a  $2\sigma$  confidence interval". 2) The 1-sigma level, instead, it refers to the cosmogenic measurements. Most of the AMS results are reported and provided to the final user with an uncertainty that corresponds to 1-sigma; therefore we used the same confidence level in our examples. We will state this differences more clearly in the text.

- I would like to see the equations that are used to do the actual blank correction (do you directly consider 9/10 ratios or the number of 10Be atoms as this may yield slightly different results if the amount of carrier that was used is not constant across all measurements).

We have reported in the text that "To perform a blank correction, the number of 10Be atoms contained in the blank is subtracted from the number of 10Be atoms contained in the sample". This implies that we have first calculated the number of 10Be atoms in samples and blanks and then done the subtraction. We will add the subtraction equations if this can help the reader understand the procedure we followed. Also, following a comment of Referee #1, we will slightly change this

sentence explaining that this is one possible approach to follow (exactly because the amount of carrier added to the samples, although very similar, is not always the exact same number for all the blanks and samples), but that other approaches (i.e., a direct subtraction of the ratios) are also sometimes used. We will explain the differences and consequences.

- There is no discussion about the low uncertainty that are associated with blank measurements and how that affects the correction. Also, how 10B isobaric interferences are corrected for in the case of low count blank determinations is something that should be mentioned and discussed.

This is not exactly correct. We have explained in section 3.3 how we treat the uncertainties associated with AMS measurements of blanks and samples. These are used for error propagation in case of the single blank correction method, whereas we used the standard errors of the mean and of the median when we used an average blank value (see equations in section 3.3). This approach allows us to include the effect of the uncertainties directly into the blank correction procedure. In addition to this, we have discussed the effects of the different uncertainties on the results in section 4.3.

For the aim and the target audience of our paper, we have designed it to not be overly technical. We want the focus of the paper to be on the statistical method, rather than on the chemical procedure necessary for Be-extraction (for which many technical and excellent papers already exist). However, we understand that such discussion on B-interference can be an important point when dealing with low concentration samples. We already have a full paragraph in the supplementary material that deals with Be contamination (Section S5). We will implement the discussion about B interference within this section, giving some references that can be useful for the readers that want to deepen their knowledge on the topic.

- A broader discussion that includes the whole preparation and measurement procedure would for instance investigate: o the impact of reducing carrier to quartz ratios in terms of overall uncertainty and blank assessment as this increases 10/9 ratios but decreases measurement time or 9Be currents ... o the influence of isobaric interferences (10B) on the low ratio measurements ... o the relative contribution of the blank correction method to the overall uncertainty and reproducibility of low

10Be concentration measurements (for instance comparing it to CRONUS or internal standard measurements) o What is the impact of very low 10/9 Be ratio carriers on the final measurement (i.e. if the variability is still as high as for commercial, higher ratio carriers).

We agree with the Referee, and indeed we reported some of this information in the supplementary material. However, because this is not the focus of the paper, we prefer to leave this information in the supplementary file and not within the main manuscript, where we believe it would distract from the focus of the paper, which is the statistical method proposed for establishing a lower threshold for interpretation of the data.

**I hope this helps to further improve the manuscript.**

Thanks for the comments, we hope to have clarified the aim and goal of our work and why we believe it will be important to have it published in a broad journal such as ESurf.

---

## Referee Comment (RC3) · Anonymous Referee #1 · 30 Jun 2017

I have read the responses of the authors to both reviews. Their response reads more as a defense of the manuscript "as is" than an acknowledgment that there are ways in which the manuscript could be improved so as to better serve the broader community.

In particular, the suggestion that the authors don't wish the manuscript to be overly technical and thus that they will only consider boron isobaric interferences in the supplement is indicative of why this paper is not ready for publication. This weakness was pointed out by both reviewers as a critical omission - one that can be a major player in the blank. Yet. rather than address this weakness, head on, they have suggested it's only important for technical readers - those just interested in results need not consider it. But, everyone interested in interpreting low ratio data needs to be fully aware that isobaric interferences are important. The similar rejection of the comments on Type 1

vs Type 2 errors - rather than clarification in the ms is another example.

The authors's defensive rather approach in responding to constructive criticism of their work by two different reviewers confirms to me that they are insufficiently self-critical and are not responding to community concerns. The responses do not change my overall opinion of the manuscript which I continue to believe is not a good fit for Earth Surface Dynamics and which I do not believe will have a wide readership. I continue to believe that this ms is incomplete, does not represent the state of the art in blank correction, could lead readers down an erroneous path, and belongs in a more specialist journal because it does not deal directly with material of interest to those whose focus is surface processes.

---

## Referee Comment (RC4) · Anonymous Referee #1 · 28 Jul 2017

I have read the revisions proposed by the authors and find them to be superficial and cosmetic rather than a detailed rethinking of the paper and its approach. In particular, the suggestions of the last paragraph reflect the naive and simple approach of the paper. These suggestions are in the literature and should have been known to the authors and referenced. See: Frankel et al., 2010, EOS and posting to Greg Balco's blog dated March, 2010. This paper needs real, deep and considered revision before it is appropriate for publication.

I continue to feel that the paper is inappropriate for ESD.

---

## Referee Comment (RC5) · D. Granger (Referee) · 28 Jul 2017

While every AMS laboratory by necessity has a protocol for dealing with low-level backgrounds and uncertainties, and most AMS laboratories have published their background correction procedures in the technical literature, the cosmogenic nuclide field in general does not have a comprehensive paper on the treatment of very low level samples. I was hoping that this manuscript would fill that void. However, it almost completely ignores a substantial body of literature on blank subtraction, in particular for Poisson processes, and takes an oversimplified approach to backgrounds and blank subtraction. I do not think that it advances the field in a significant and widely useful way.

As explained, for example, by Elmore et al. (1984) and many others [e.g., Donahue et al. 1990; Wacker et al., 2010; ...], the blank or background in an AMS measurement depends on (1) contamination during sample preparation; (2) contamination in the ion source, commonly referred to as 'cross-talk'; and (3) tails of other ionic species or isobaric interferences in the detector, commonly referred to as 'interference'. In addition, the blank can also include contamination in the cathode material (Middleton et al., 1994), and the metal binder mixed with the sample, here subsumed under sample preparation. These effects are listed in the supplementary information section S5 but should rightfully belong in the main text due to their importance.

Treatment of the background depends on the source of the counts. If the counts are due to consistent laboratory contamination from reagents, then it is appropriate to treat the blank as a fixed number of atoms added during sample preparation. In this case, the number of atoms added to the sample during preparation is calculated and then subtracted from the total number of atoms determined for that sample, as is done in this manuscript. If the background is due instead to cross-talk, then the blank should more appropriately be subtracted as a count rate. This is because the background 10Be ions are being evaporated from various surfaces inside the source at a constant rate, independent of sample beam current. Finally, if the background is due to interference in the detector (most likely from boron, unless B interference is eliminated by a gas-filled-magnet or by post-stripping), then there is a correction factor applied that is proportional to the interference count rate, determined empirically for each laboratory and most often for each run. Each AMS laboratory should evaluate the sources of background and provide this information to the user whether they make the 'machine' background subtractions or not.

Relevant to machine backgrounds, Savi et al. mistakenly state that a cross-talk value of 0.1 permil implies that this source of background can be ignored. Unfortunately this is not usually the case for low-level blanks. AMS standards are typically much higher ratio than the samples, so that the standardization can be made with high precision.

**ESurfD**
This manuscript did not state where the measurements were made, but the Savi et al. (2016) paper from which the data were derived states that they were made at the University of Cologne against standards with 10Be/9Be ratios of 5.35 x 10-13, 6.36 x 10-12, and 2.71 x 10-11. Dewald et al. (2013) indicate a beam current of 5  $\mu$ A for beryllium and a transmission of 33% at the Cologne AMS. That would correspond to roughly 5 counts per second (cps) at the detector for the lowest standard, and about 250 cps for the highest standard. A cross-talk value of 1 x 10-4 would then correspond to a count rate of one count per 30 minutes for the lowest standard and about 1-2 cps for the highest standard. Is this really negligible? Savi et al. don't state their beam currents for blanks and samples, but blank 10Be/9Be ratios range from about 1E-16 to 1E-15. If the blanks and samples have the same current as the standards (which they usually don't), then a blank of 1E-16 would correspond to about 1 count per 16 minutes. At this level, even if only the lowest standard were previously run, cross-talk would account for around 1/3 of all counts. If higher standards were used instead, or if higher ratio samples were previously run in the source, then cross-talk could completely dominate the measured AMS background. It is unclear how much of the variation in the blank, then, is due to laboratory cleanliness versus cross-talk in the AMS source for low-level blanks without additional information such as machine blanks that were run concurrently. Certainly if the higher ratio samples (e.g. 53-57) were run adjacent to a blank they would contribute on the order of 1 count per few minutes, or up to 5-10 counts due to cross-talk, depending on the run time.

Making the problem more confusing, there may very well be idiosyncrasies and statistical overestimates of the blank at various laboratories. I know from personal experience that at PRIME Lab and at LLNL a zero blank is never reported. If no counts are detected, then at the end of the run a single count is added artificially and assigned an uncertainty of 100%. This is a conservative approach that overestimates the true value of the blank. (Four of the blanks in this paper have 100% uncertainty, suggestive of a single count. Most have between two and twenty counts in the detector, based on the stated uncertainties.) In addition, the machine blank may easily be variable from run-

**ESurfD**
to-run at the AMS facility, depending on variables such as cross-talk (which changes over time during a run), the AMS tune, and the defined region of interest in a dE/dx detector.

The most important oversimplification in this paper, and the one that is the most concerning, is the use of Gaussian error analysis. An AMS measurement divides nuclear counts in a detector by the beam current measured in a Faraday cup. The number of counts is a Poisson process governed by both time and beam current, and \*must\* be treated using Poisson statistics. This is especially critical for blanks and low-level samples, where Poisson statistics deviate strongly from Gaussian (e.g., Currie, 1972). There is in fact a robust literature on low-level Poisson statistics and measurement backgrounds for radioactivity counting, particularly related to health physics. I would argue that if you want to distinguish a low sample from a blank, then the correct approach would be to compare either the counts per Coulomb or counts per time using net Poisson statistics, with the difference being whether you consider the background to come from cross-talk or from intrinsic 10Be in the blank. Methods should follow. for example, Potter and Strzelczyk (2008; 2011), Alvarez (2013), or others dealing with low numbers of counts and variable counting times. Alternatively, a Bayesian approach modified from examples such as Mathews and Gerts (2008) and references therein could be used. The point is that there is an entire literature on blank subtraction and detection limits using Poisson statistics that is almost completely ignored in this manuscript.

Admittedly, for high blanks and samples there are more than enough counts in the detector to justify the use of Gaussian uncertainty analysis, and some AMS laboratories follow this sort of background subtraction (e.g., Nadeau and Grootes, 2013). However, as the authors show, the distribution of blanks is better described by a binomial distribution and a confidence interval or likelihood approach would be preferred. In the end, the treatment of high blanks and high ratio samples, though, is just not as interesting a problem as the treatment of low-level samples and blanks, it is one that is more easily

**ESurfD**
resolved, and it has already been dealt with in numerous technical AMS papers. The real advance that could be made in this paper would be to show how to properly deal with low-level samples that have measurements near the blank.

I appreciate the discussion in the manuscript about whether to choose a smaller number of blanks corresponding to chemistry batches or a longer-term average blank value. The philosophical approach is very different, depending on what one believes the source of the blank to be. I tend to favor more of a time series approach when evaluating my own blanks relative to known 'machine' blanks or unprocessed carrier material. If my blank is substantially higher than the machine blank then I know that my samples have been contaminated in the lab. If the unprocessed carrier blank is high, then that indicates a problem with the AMS measurement such as cross-talk. The blanks reported here do show some patterns over time, with significantly higher values starting at number 36. Is that due to laboratory contamination or AMS conditions? It is not at all clear that the later blanks should be combined statistically with the earlier blanks. Before doing so there should be some sort of time series analysis to show that there is no drift or trend.

Some smaller issues arise in the paper that should be dealt with in any revision or re-submission. (1) AMS is drastically over-simplified in the description in section 2.2. (2) There seems to be some confusion about the presence or absence of beryllium in minerals. Beryllium is almost universally absent in quartz, but variably present in most other minerals at the ppm level.

In summary, I would very much like to see a paper that goes through how to deal with low-level measurements in a thorough manner, and makes strong recommendations based on robust statistical arguments about the proper way to make background corrections. This paper presents several different options but never really gets into low-count statistics and sweeps several important issues under the rug. While it does address interesting problems of detection limits, I don't see that it brings the field forward in a general way beyond essentially normal practice.

**ESurfD**
**-Darryl Granger**

References cited in addition to those in the manuscript:

Alvarez, J.L., 2013. Correction to the count-rate detection limit and sample/blank timeallocation methods. Nuclear Instruments and Methods in Physics Research Section A: Accelerators, Spectrometers, Detectors and Associated Equipment, 729, pp.725-727.

Dewald, A., Heinze, S., Jolie, J., Zilges, A., Dunai, T., Rethemeyer, J., Melles, M., Staubwasser, M., Kuczewski, B., Richter, J. and Radtke, U., 2013. CologneAMS, a dedicated center for accelerator mass spectrometry in Germany. Nuclear Instruments and Methods in Physics Research Section B: Beam Interactions with Materials and Atoms, 294, pp.18-23.

Donahue, D.J., Linick, T.W. and Jull, A.J.T., 1990. Isotope-ratio and background corrections for accelerator mass spectrometry radiocarbon measurements. Radiocarbon, 32(2), pp.135-142.

Elmore, D., Conard, N., Kubik, P.W. and Fabryka-Martin, J., 1984. Computer controlled isotope ratio measurements and data analysis. Nuclear Instruments and Methods in Physics Research Section B: Beam Interactions with Materials and Atoms, 5(2), pp.233-237.

Mathews, K. and Gerts, D., 2008. Bayesian analysis for very-low-background counting of short-lived isotopes: Lowest minimum detectable quantity. Journal of Radioanalytical and Nuclear Chemistry, 276(2), pp.305-312.

Nadeau, M.J. and Grootes, P.M., 2013. Calculation of the compounded uncertainty of 14 C AMS measurements. Nuclear Instruments and Methods in Physics Research Section B: Beam Interactions with Materials and Atoms, 294, pp.420-425.

Potter, W. and Strzelczyk, J., 2008. Computer code for detection limits and type II errors with unequal sample and blank counting times. Journal of Radioanalytical and Nuclear Chemistry, 276(2), pp.313-316.

**ESurfD**
Potter, W.E. and Strzelczyk, J.J., 2011. Improved confidence intervals when the sample is counted an integer times longer than the blank. Health physics, 100(5), pp.S67-S70.

Wacker, L., Christl, M. and Synal, H.A., 2010. Bats: a new tool for AMS data reduction. Nuclear Instruments and Methods in Physics Research Section B: Beam Interactions with Materials and Atoms, 268(7), pp.976-979.

**ESurfD**

---

## Author Comment (AC3) · 28 Jul 2017

In this revised version of our paper we have addressed most of the critical points raised by Referee #1. In particular, we hope to have clarified the aim and the target audience of our work, which we believe were the first cause of misunderstanding. Following the most constructive comments of the referee, we have:

- Introduced a brief discussion on boron interference and how to deal with it; - Explained the differences between type I and type II errors in statistics, and how it could influence the choice of the confidence interval for the definition of a lower threshold; - Added more references, including the papers suggested by the referee.

We have also analyzed in details other papers that have dealt with low cosmogenic

concentrations, as suggested by the referee. Unfortunately, due to missing information in the original manuscripts, we have not been able to use these papers in our examples (see the last point in the detailed answer for more information on this issue).

This latter point highlights once more the importance of our work, and the necessity to raise the awareness of people dealing with low cosmogenic concentrations on the problems and the issues that should be considered during the whole research, from sample preparation to the interpretation of the results. Also, a common reference on how to report and interpret data is necessary to make results comparable and re-usable. We hope to have clarified our position, and why we believe this paper is valuable, timely, and of broad interest for the cosmogenic nuclide community.

Please also note the supplement to this comment:
https://www.earth-surf-dynam-discuss.net/esurf-2017-30/esurf-2017-30-AC3-supplement.pdf

**Supplement:**

*Answer to Anonymous Referee #1: with respect to the Reviewer's comments, both in the first review and in the answer to our answer, we believe that most of the criticisms raised by the Reviewer lie in the misunderstanding of the aim and the target audience of our paper. Our "defensive" answer does not mean that we are not accepting criticisms or suggestions on how to improve the paper (which we would always accept when constructive). It simply means that we think that some of the suggested changes will not bring benefit nor clarifications to the paper. Nonetheless, we have made many of the changes that the Reviewer suggested, and explained why we do not believe that some of the suggested changes are necessary for the goal and readership of our manuscript.*

*In the text that follows, we have continued the answer to the Referee's first comments. Since we finally did change some things that the Referee was pointing out in the first comment, we have highlighted the new changes in the first part of the answer in a darker underlined color and we have written them in Italic.*

*Summary comments*: Thanks for the detailed review. We understand the perspective of Referee #1 with respect to the technicality of the paper and its broader interest for the surface processes community. However we don't agree completely with his/her assertions.

We would like to highlight that in this paper we describe a statistical method that defines a lower threshold useful for the interpretation of low-concentration samples. This method is known in analytical chemistry but has never been applied to cosmogenic studies. The method is designed to be used by the final user, after the chemical procedure and the AMS measurements of the samples, and it is not simply an "*exercise done to understand the potential limits of one laboratory*". Rather it is designed to help the cosmogenic nuclide community users define the geomorphic meaning of samples with low nuclide concentration. In no way we suggest that the results which do not lie significantly below the blank threshold cannot be used, but rather we use this threshold to determine whether or not a rate or age can be quantified or when we can only interpret the data in terms of a maximum age or minimum erosion rate.

In the following lines we will proceed with commenting the main concerns of Referee #1 (in black ink), especially considering the aim of our work and our targeted audience, which define the

structure and meaning of our paper. A more detailed answer on the line-by-line comments and changes to the manuscript will follow in the next weeks.

**Anonymous Referee #1**: Summary: This ms applies what are considered standard approaches in analytical chemistry to determining detection limits for the analysis of cosmogenic 10Be. The ms uses a long term set of blanks and sample measurements from one lab but multiple operators to make calculations and determine several statistical parameters for the detection of 10Be at levels confidently above the blank.

The manuscript does not break new ground; it is an exercise done to understand the potential limits of one laboratory much more than a significant advance in cosmogenic nuclide science and therefore, not the type or style of paper for publication in a broad readership journal such as Esurf. Overall, this paper presents a data set that most cosmogenic labs have and have likely analyzed internally but the ms does little to advance cosmogenic nuclide science more broadly. As a referee and as a user of this literature, I consider the ms to be much more appropriate for an AMS-specific journal such as NIMs and as such suggest it be shortened and submitted for publication there rather than eSurf. It just does not fit well nor will be it be of interest to most of the surface process community. It is technical in nature and does not have any significant geomorphic impact.

*Significance and Readership*: First of all, we would like to highlight that the aim of this paper is to address the issues that one encounters with low 10Be content samples (i.e., $< 10^5$ atoms) and to provide a standardized method for the interpretation of the results. The paper is designed for the cosmogenic nuclide final users, including all the people who use cosmogenic nuclides to study earth-surface processes, but who are not necessarily familiar with the technicality of the Be-extraction/measurement procedure. In the last decades, the rapid expansion in the use of this technique has created a growing community of users that apply cosmogenic studies without the detailed technical knowledge typical of the people who work in an institute where it is possible to perform Be-extraction and/or AMS measurements (e.g., the LLNL facility) and who may have only a basic knowledge of the technical aspects of these procedures. This paper aims also at this broader community and, as such, it is specifically designed not to be technical, because we want it to be suitable for non-technical users. Some technical information on the chemical procedure for quartz extraction, Be-isolation, and the possible sources of blank contamination are, however,

additionally provided in the supplementary material. *Considering the importance of boron interference for samples with low 10Be content, we have added a sentence and some reference also in the main text, so that the readers can be aware of the necessary precautions that can be taken when dealing with these kind of samples.* Other detailed information on very specific technical aspects, such background AMS beam currents and blank counts, are already central topics of existing and excellent technical papers that the readers can access, if interested (we provide some recent summary papers as examples, e.g., Balco, 2011; Granger et al., 2013; and some technical papers that can be of interest for the more curious users, e.g., Schaefer et al., 2009; Rood et al., 2010; Merchel et al., 2013; Portenga et al., 2015; Corbett et al., 2016). Most of this technical information, indeed, is available for internal usage of specific laboratories (e.g., to define the precision of the AMS machine and/or cleanliness of the laboratory) but are normally not included in the final AMS measurement report that is delivered to the users.

For the above-mentioned reasons, our paper does not deal with technical aspects of the AMS measurements (for which, we believe, more suitable and technical papers already exist), but rather proposes a statistical procedure (not chemical nor isotopic) based on clear definitions of detection limits, which can be used as standard reference in the interpretation of cosmogenic nuclide measurements. As such, the paper targets a broad audience and it describes a method that can be applied by all cosmogenic nuclide users. Specifically, the method provides a statistical threshold to evaluate whether the nuclide concentration can be used to quantify an exposure age or erosion rate versus only limit the age or rate; in the latter case, the data is not rejected (nor there is the risk to *reject data that are likely real*); on the contrary, a limiting value can still be very helpful. This proposed method, additionally, can also be used in cases where low-concentration samples are not expected. In such cases, the user may not be aware of the precautions needed during the chemical procedure, but he/she will still need to define how the results should be interpreted. For this goal, the paper describes a procedure that is commonly used in analytical chemistry to define a statistically significant threshold for low-concentration samples and provides a method for its application in the field of cosmogenic nuclide studies. Consequently, it does not describe a "new procedure", but rather we propose to apply this standardized method for cosmogenic nuclide studies, seeing as such a procedure is so far missing in the community. This aspect is very relevant, especially considering that a growing number of researchers will have to deal with low

concentration samples (see Corbett et al., 2016). We believe that a common and standardized approach that defines a general rule for the interpretation of these data is timely and valuable, and that it can be of interest for a broad audience rather than only for the readers of a technical journal such as NIMs. For these reasons, we strongly believe that the paper will fit well a journal such as ESurf.

Furthermore, the ms is proscriptive and narrow in its approach, which makes it less likely to be accepted by the community. The manuscript does not set the presented data in context because it does not include a critical evaluation of how previous workers have done blank corrections nor does it demonstrate how different blank corrections would change geomorphic outcomes of extant studies. There are several examples in the recent literature where workers have applied several different approaches to blank correction and tested the sensitivity of results to varying approaches; see for example Corbett et al., GRL (2017) on 10/26 ratios. These are not cited nor are they considered critically. The ms reviewed here is under-referenced, omitting numerous important citations both classical and recent that are germane.

*As suggested from the referee, we have added in paragraph 2.3.1 a longer description of how other people have dealt with blank correction. We added the reference to Corbett et al (2017) in several points of the manuscript. This paper is, to our knowledge, the only published paper that discusses different blank corrections. They discussed it, however, only in terms of blank statistics, not highlighting how the choice of a different approach would change the 10Be atoms in the samples or the geomorphic interpretation of a sample (this is also due to the fact that their statistics are very similar, so that differences between the various mentioned approaches would be negligible).*

With respect to the blank subtraction methods, we have presented here three different approaches and discussed how their application would change the final interpretation of the results. So we do not agree with Referee #1 when he/she said that we did not consider different blank subtraction methods or that we did not "*demonstrate how different blank corrections would change geomorphic outcomes of extant studies*". Nevertheless, in the revised version of the manuscript, we have added more information on how other people have dealt with this topic (e.g., Corbett et al., as suggested by Referee #1) and discuss their approach. We would also like to highlight that very few papers discussed and reported in detail how the blank correction has been performed, and

even fewer papers additionally dealt with low 10Be content samples (i.e., $< 10^5$ $^{10}$Be atoms). As such, this information is not so easy to find within the existing literature.

This point raised from Referee #1 demonstrates how a common procedure for these important steps in the calculation of the cosmogenic results is still missing within the community. The way people normally deal with blank corrections is often not reported in the papers, but this could lead to slightly different results and to a lack of comparability between different publications. This highlights once more why our paper will represent an important contribution for the cosmogenic nuclide community.

Also, we have already provided, as examples, some of the most recent summary papers (e.g., Balco, 2011; Granger et al., 2013) that provide an exhaustive explanation of the cosmogenic nuclide technique and its latest advances. Interested readers can go into the details of the references mentioned within these papers, based on their own specific interests. In the revised version of the manuscript, we will add more references to recent papers (e.g., Corbett et al., 2017), as suggested by Referee #1.

Critically, the paper does not consider type 1 vs type 2 errors, that is in striving to be certain that 10Be is confidently detected, such as using 99.9% confidence, samples containing real 10Be are almost certainly being rejected. This will lead to errant science and must be considered head on in any revision before publication. It is a critical flaw and must be addressed before publication in any journal.

*For completeness of the paper, we have added the theoretical description of these two types of error in section 2.4.1. However, we would like to highlight that this information would not change the results of the proposed approach. It is true that depending on the chosen confidence level the values of the two errors would change, and is quite obvious that with a lower confidence interval more samples could be used for a quantitative interpretation of ages or rates (i.e., the probability of the type II error would be reduced). However, decreasing the confidence interval would also increase the risk of using a sample for a quantitative interpretation when in reality the 10Be content of that sample may be in large part derived from laboratory contamination (the probability of the type I error would increase). In our opinion, accepting a higher risk of incurring in a higher*

*type I error (thus reducing the confidence interval) is worse than accepting the risk of a higher type II error, and this is one reason for which we propose to use the LOD value as lower threshold for cosmogenic studies. We have better specified this point in the paper.*

*In this paper we describe an existing approach which uses high confidence intervals, and we explain how to apply this approach to cosmogenic studies. We have reported the definitions of LOD and LOQ used in analytical chemistry (respectively representative of 99.9% and 99.9999% confidence intervals) and the users should be aware that changing the confidence interval would alter these definitions. Nonetheless, we have stated in the discussion and in the conclusion that whenever a user want to use a lower confidence interval (e.g., 2σ), the important point is to report it in the paper, together with all the necessary information that would make the results of that study comparable with others.*

*Additionally, we would like to highlight that despite the chosen confidence interval, in no way "a sample containing real 10Be" would be rejected, as with this approach all the samples can be used. What changes is only the way with which we can interpret the geomorphic meaning of the samples (e.g., for the quantification of an age or a rate, or as limiting number, a still very helpful and meaningful result).*

Lastly, boron, an isobaric interference is neglected. It varies sample to sample and the means by which it is rejected varies between AMS facilities. It can be a very real component of the blank measurements. At minimum, it needs to be presented and discussed in the context of the ms and the measurements within – better yet would be to consider it broadly across the community. Again, this is narrow, technical information of interest to a small section of the community but critical to the issues here.

We agree that this is a "narrow, technical information of interest to a small section of the community" and for the reasons already mentioned above, we did not go into the detail of this discussion. Additionally, as pointed out by the referee, this is an aspect that changes from AMS to AMS laboratories and its discussion goes far outside the aim of our paper and our targeted audience. Nevertheless, *we have added a sentence to acknowledge the importance of this aspect*

*when dealing with low-concentration samples and reported examples on the possible procedures to use in this case, with the related references for the interested users.*

In summary, the ms is not appropriate for Esurf, is too narrowly focused on one approach to blank subtraction, ignores Type 1 vs Type 2 errors, does not consider 10B interference, and has little critical evaluation of the literature in which the current data need be considered in context. As is, it is a formulaic approach to analyzing an isotopic rather than geomorphic data set and not a significant advance of geomorphic science.

We have presented here three types of blank subtraction, an aspect that is not normally discussed in cosmogenic papers. We have provided additional information using the work of Corbett et al. to include other possible approaches to blank corrections. We have explained why our paper does not want to be too technical (still, technical information can be found in the supplementary material), considering the aim and the target audience for this work, and we have provided two examples of how our proposed approach can be applied to geomorphic studies, discussing its consequences for the geomorphic interpretation of the samples (section 5.4). In the revised version of the paper we will include more examples, as suggested by Referee #1, so that the readers can have a broader idea of the impact of the proposed method on geomorphic studies.

For all these reasons, we believe that the paper is of broad interest and that it presents timely and necessary information for the cosmogenic nuclide community and for a correct interpretation of low-concentration samples. Its publication in a broad journal such as ESurf, rather than in a technical journal, will surely be of benefit for many researchers within and outside the cosmogenic nuclide community.

Suggestions for improvement:

I would encourage authors to expand their selection of references in particular citing more studies that make very low level measurements and how these studies have dealt with blanks as well to cite some of the many excellent review papers since 2010 that compile both erosion rate and

exposure age data. For example, Carlson and Nelson have recently both published very low concentration measurements from Greenland, one for glacial dating and the other for sediment tracing and neither of these studies are cited.

*Thanks for this suggestion. We have added the two papers mentioned by the Referee plus some*
5    *other study. Two of the "excellent review papers since 2010" (e.g., Balco, 2011; Granger et al., 2013) have already been cited in the manuscript as examples and more detailed literature can be found within these papers.*

P2, Ln 25: most AMS measurements in the 10-15 and 10-16 range are not very precise, reword.

10    *We have reworded the sentence.*

P2, Ln 27: This would be an appropriate place to cite Corbett et al. (2016) who review in great detail lab and AMS issues affecting detection including blanks and AMS beam currents. They in turn cites others such as the work on precision by Rood at the LLNL facility.

15    *We have added the reference; a direct citation to Rood et al. is already provided on line 28.*

P2, Ln 30-35: It is not clear why a standardized approach is needed or an improvement on the current approach; there is so much buried in blanks including which AMS, operator changes, real contamination. The paper would likely be better accepted by the community if it were to provide
20    a means or variety of means by which the blank subtraction could be done. The way this section is written presumes the authors have defined "the" way to do blank correction not "a" way to do blank correction. This will not advance the science of AMS and 10Be.

*We believe proposing a standardized approach to blank correction in cosmogenic nuclide studies*
*that follows well established procedures in the field of analytical chemistry is important to have*
25    *comparable data and to give the users a reference approach that can be used to interpret the geomorphic meaning of low concentration samples. This is the aim of this paper.*

*The "variety of means by which the blank subtraction could be done" has been added to section 2.3.1 where it better fits within the content.*

P3, Ln 10: This sentence is incorrect. 10Be IS naturally present in earth materials.

*We have changed the sentence.*

P3, Ln 17:  This set of references fails to cite the original 3 references for nuclides in sediments – a critical oversight that needs to be remedied.

*We have added the missing references.*

P4,  Ln1:  This sentence omits an important part of running blanks,  determining the dark current or background of the AMS system including cross talk in the source. This needs to be mentioned and cited properly.

*This is a very technical comment, and touches on issues that the final user of AMS data is in no way involved with. We are not attempting to identify all sources of uncertainty in blanks, but rather we are suggesting a procedure for how end-users can determine if they may place uncertainty limits on their exposure ages or erosion rates, or if they can simply limit the age or rate.*

P4, Ln 5-8: This is at least not correct for our lab and I think not correct for other labs. For us, every sample including blanks gets the same reagent amount and same open beaker time. Otherwise, how could we could compare process blanks and samples and do the subtraction in a meaningful way?

*In our experience samples with different amount of quartz always require different amounts of acid and different time periods of open backers for the dissolution of the samples. Nonetheless, we have changed the sentence to include different possible procedures.*

P4, ln 15-20: This section omits a critical issue in defining blanks on the AMS – boron as an isobaric interference and how that is handled. The process varies between AMS (some such as PRIME use GFM that completely removes 10B, others like LLNL make a very uncertain correction, others use post stripping). For low level samples, this is critical.

*We have acknowledge the issue in the text.*

P4, ln 21-22: This is one way to do it but if the same amount of carrier is added to all samples, then the ratios can be subtracted. This alternative approach needs to be acknowledged and cited.

*We have changed the text.*

P4, ln 36: there needs to be more here. How do blanks change over time? Does contamination increase over time in labware?

*As we have mentioned in the text, "this is a point that needs to be addressed for each laboratory and perhaps at the individual-user level".*

P5, Ln 7-10: This is far too simplistic. Only considering the upper value of the blank will result is rejecting data that are likely real. The blank subtraction process is a probabilistic one and different for different purposes. The goal of determining whether something is confidently detected is very different than the goal of best estimating the blank for subtraction. The paper would be much stronger if it considered this subtlety.

*As stated in previous responses, we have tried to clarify that when a sample yields a 10Be content that cannot be distinguished from the blank, the data does not need to be rejected as meaningless. This is an important point, and the referee's confusion on it highlights the need for this point to be clarified to the community. The maximum 10Be content of the blank distribution can still be interpreted in terms of an exposure age or an erosion rate; when a sample's 10Be content falls below this threshold, we can still say that it has a maximum age or a minimum rate equal to that calculated for the upper limit of the blank distribution.*

P6, Ln 4: Unmentioned here is the fact that blanks are very imprecise measurements by their nature and because of the Poisson distributed counting statistics of AMS. The lowest blanks can contain only a count or two (or even none). Implicit in the discussion above is that the blanks are normally distributed (parametric statistics). If these issues were discussed, the paper would be much stronger. Some of this discussion is in section 3 and could be moved up.

*Indeed, this is an important aspect of the paper and it is exactly why we discuss the possibility to use a different approach than the one recommended by the IUPAC, which assume a normal distribution. We clearly stated this issue at the beginning of Section 3 (only 4 lines below the mentioned point of Referee#1) where we say that "This recommendation [intended as the IUPAC recommendation], however, assumes a normal distribution of values, which is rarely the case when dealing with low concentrations of an analyte" and explain later in section 4 why this is important and how it changes the results of the statistical interpretation. This is a critical aspect of the proposed method that is already discussed through the paper.*

P8, Ln 5: "Although a minimum of 20 values" this has been stated repeatedly. No need to state again.

*Thanks for this comment. We have noted that we have repeated this sentence often through the paper. However, this sentence in this position helps us to explain why we use two different sets of blanks despite having a low number of blanks (n=8). We have removed this repetition from other parts of the text.*

P8, Ln 35: It is not clear how the average blank constrains any temporal variance? Please explain.

*The average blank value constrains the temporal variance because the blanks are created in a relatively short period of time. This is explained few lines above this point where we stated that "In the average-blank correction, all the blanks processed in multiple batches by one operator over a limited time frame […] are used to obtain a representative value of 10Be atoms for the blanks […]".*

P9, ln 18: I find this paragraph very hard to follow and not very informative. A table or graphic would convey the same information much more effectively. Some of the information is summarized in tables but not all of it.

*We have created a new table (new Table 3) and wrote all these values there.*

P11, ln 13: This is a critical mis-understanding of Type 1 and Type 2 errors. Here, the authors suggest that, "In general, the use of the long-term laboratory blanks (being based on many blank measurements) guarantees more reliable values for the statistics of the blank distribution and for the calculation of the determination limits; as such, they may be preferred." The approach the authors suggest is very likely to introduce errors of rejection for data (samples) that contain actual 10Be above blank levels.

*This is a critical aspect of the paper that we have focused on throughout the text, in the figures and tables. It is not a misunderstanding of the two types of statistical error, but rather the description of how the choice of the blank distribution to use for the blank correction may change and affect the interpretation of the results. And again, no data will be rejected, only a higher number of samples can only be used as limiting erosion rates or exposure ages, as we show in the manuscript. This is exactly why it is important to choose the correct method for the blank subtraction and for the calculation of the threshold, and is one of the main conclusion of the paper. We have demonstrated through the paper that these elements will strongly change the results of the statistical approach and the way people can use and interpret the data.*

*Also, we have stated in the following line that "Nevertheless, when the long-term blank ensemble shows a large variance, the assumption of unchanging laboratory conditions is unlikely to be valid, and the blank measurements are unlikely to be representative of the variation occurring within a single batch. Under these circumstances, and when there is an acceptable number of blank measurements available (at least 20; Bernal, 2014), a set of blanks obtained from a single operator over a shorter time interval may be favored for the calculation of the threshold".*

*We are discussing here the results of our proposed work, differentiating between the several approaches that we describe through the paper. To report only one sentence and say that we are suggesting something wrong, omitting all the other part of the discussion, would give an over-negative impression of our work and of the detailed analyses we have done.*

Section 5.4: This is not an adequate critical review of what others have done with low activity samples. The data tables for AMS would be much more informative if they included the standard(s) to which the ratios were normalized, more about the boron counts and rejection procedures, comparison of sample beam currents to standard beam currents, the number of gated 10Be counts, and the actual uncertainties of the measured ratios.

*We have included into table 2 the standard to which the ratios were normalized and the amount of added carrier. The AMS uncertainties associated with the measured ratios are already included in the table, whereas the other information are not in our possession, since they have not been included in the AMS reports or were not reported within the publications.*

15 *Also, we have carefully read some of the papers that the Referee suggested as examples for publications with low-concentration samples. Unfortunately, due to missing information about sample or blank values in those publications, we were not able to use them as examples in our discussion on the "Implication for geomorphic applications".*

*In particular, the paper of Nelson et al (2014) report the average blank ratio used for the blank correction but not the 9Be amount added in the carrier nor the carrier concentration. Considering that we need to calculate the LOD and LOQ values using the 10Be atoms included in the blanks, without this information, we cannot convert the blank ratio into 10Be atoms and thus we cannot calculate the desired thresholds.*

20

*The paper of Corbett et al. (2017) reports data form previous publications. Looking in detail into these other papers (Corbett et al., 2011 and 2013), we have found that the AMS ratios for the samples have only been reported after the blank correction. The raw AMS results for the samples are missing. Additionally, these authors performed the blank correction using two different averaged blank ratios (one derived from commercial blanks and one derived from beryl blanks). Not knowing which samples have been corrected with which blank value, we cannot re-calculate*

25

*the samples' ratios before the blank correction. So also for that paper we are unable to calculate the 10Be content of the samples.*

*In the paper of Carlson et al. (2014) all the necessary information for the blanks are reported, but unfortunately all of the AMS data for the samples are missing.*

5  *As we have already mentioned above, "very few papers discussed and reported in detail how the blank correction has been performed, and even fewer papers additionally dealt with samples with low 10Be content (i.e., < $10^5$ 10Be atoms). As such, this information is not so easy to find within the existing literature. This highlights once more why our paper is worth to be published and why it will be an important contribution for the cosmogenic nuclide community.*

---

## Author Comment (AC4) · 28 Jul 2017

**Determination limits for cosmogenic [10]Be and their importance for geomorphic applications**

Sara Savi[1], Stefanie Tofelde[1,2], Hella Wittmann[2], Fabiana Castino[1] and Taylor F. Schildgen[1,2]

[1] Institute of Earth and Environmental Science, University of Potsdam, Karl-Liebknecht-Str. 24, Haus 27, 14476 Potsdam-Golm, Germany
[2] Helmholtz Centre Potsdam, GFZ German Research Centre for Geosciences, Telegrafenberg, 14473 Potsdam, Germany

*Correspondence to*: Sara Savi (sara.savi@geo.uni-potsdam.de)

**Abstract.** When using cosmogenic nuclides to determine exposure ages or denudation rates in rapidly evolving landscapes, challenges arise related to the small number of nuclides that have accumulated in surface materials. Improvements in accelerator mass spectrometry have enabled analysis of samples with low $^{10}$Be content ($<10^5$ atoms), such that it is timely to discuss how technical limits of nuclide determination, effects of laboratory cleanliness, and overall sample preparation quality affect lower blank limits. Here we describe an approach that defines a lower threshold above which samples with low $^{10}$Be content can be statistically distinguished from laboratory blanks. In general, this threshold depends on the chosen confidence interval. In detail, however, we show that depending on which ensemble of blank values and which approach is chosen for the calculation of this threshold, significant differences can arise with respect to when a sample can be distinguished from a blank. This in turn dictates whether the sample can be used to determine an exposure age or a denudation rate, or when it only constrains a maximum age or a minimum denudation rate. Based on a dataset of 57 samples and 61 blank measurements obtained in one laboratory, we demonstrate how these different approaches may influence the interpretation of the data.

**Copyright Statement.** The Authors agree with the Licence and Copyright Agreement of the Earth Surface Dynamics Journal.

**1 Introduction**

In the last two decades, the use of *in situ*-produced cosmogenic nuclides for the quantification of denudation processes and the determination of exposure ages of landforms has seen a rapid expansion (Balco, 2011; Granger et al., 2013). This development is due to advances in the technique and to the wide range of geological environments in which the method can be applied. Comprehensive summaries of the method can be found in Anderson et al. (1996), Bierman and Steig (1996), Granger et al. (1996), von Blanckenburg (2005), Balco (2011), and Granger et al. (2013). Among the suite of cosmogenic nuclides that can be used to study geomorphic processes (e.g., $^{10}$Be, $^{26}$Al, $^{36}$Cl, $^{3}$He, and $^{21}$Ne), *in situ*-produced $^{10}$Be is the most widely used, especially for the quantification of denudation rates (von Blanckenburg, 2005). For simplicity and clarity, we will focus our discussion on *in situ* $^{10}$Be produced in the target mineral quartz only, although similar concepts can be applied to other cosmogenic nuclides. The broad expansion of $^{10}$Be applications includes studies that extend the limits of the technique by analyzing nuclide concentrations in environments where some of the assumptions inherent to the method are not always satisfied. These studies explore, for example, landscapes that are not in erosional steady-state, e.g. due to recent glaciation (Wittmann et al., 2007), settings where different rock types do not contribute quartz equally (Safran et al., 2005; Torres Acosta et al., 2015), and environments prone to mass failures or with non-uniform sediment supply (Niemi et al., 2005; Binnie et al., 2006; Yanites et al., 2009; Kober et al., 2012; McPhillips et al., 2014; Savi et al., 2014; Schildgen et al., 2016). It is particularly challenging to apply these techniques in environments where the cosmogenic nuclide content is low. For example, the occurrence of deep-seated landslide or debris-flow events in rapidly eroding landscapes may result in admixing of low cosmogenic nuclide concentration material into fluvial sediments (Niemi et al., 2005; Yanites et al., 2009; Kober et al., 2012; Savi et al., 2014). Likewise, recently exposed bedrock surfaces contain low $^{10}$Be due to their short exposure time to cosmic rays (Licciardi et al., 2009; Schaefer et al., 2009; Schimmelpfenning et al., 2014; Nelson et al., 2014; Carlson et al., 2014; Savi et al., 2016; Corbett et al., 2017). In other cases, scarcity of the target mineral within the collected material can limit the total amount of nuclides in the sample. Difficulties encountered with low $^{10}$Be-content samples are related to the technical limits of the Accelerator Mass Spectrometer (AMS), which can precisely measure $^{10}$Be/$^{9}$Be ratios down to $10^{-15}$ or $10^{-16}$ (Stone, 1998), as well as technical limits and cleanliness issues related to the laboratory where the samples are prepared (e.g., Balco, 2011.; Corbett et al., 2016).

[revised manuscript text omitted]

*Machine* or *instrument blanks* (generally related to AMS measurements) indicate the precision at which the AMS can measure the $^{10}$Be/$^9$Be ratio (Balco, 2011). This latter kind of blank is uninfluenced by contamination that occurs during the chemical procedure in the laboratory and can provide information about the sensitivity of the measuring process. However, for samples with low $^{10}$Be content, the presence of boron can represent an important source of interference for the AMS measurements (e.g., Matsuzaki et al., 2007; Chmeleff et al., 2010; Corbett et al., 2016; Marrero et al., 2016). Since different AMS laboratories deal with this issue in different ways (e.g., $^{10}$B corrections, post-stripping, B-reduction with microwave), in the case of low $^{10}$Be content, precautions for B-removal or B-correction should be discussed with the AMS laboratory where the measurements will be done, carefully considered and undertaken. In accelerator mass spectrometry, cross-contamination due to long-term memory of the AMS measurements is in the order of 0.1‰ (Rugel et al., 2016), so that the machine background is commonly neglected (Currie, 2008). It follows that most of the contamination stems from the laboratory processing of the samples (Balco, 2011) and thus we focus only on *laboratory blanks* hereafter.

**2.3.1 Blank corrections**

The *blank correction* can be performed either by subtracting the $^{10}$Be/$^9$Be ratio of the blank from the $^{10}$Be/$^9$Be ratio of the sample, or by subtracting the number of $^{10}$Be atoms contained in the blank from the number of $^{10}$Be atoms contained in the sample. The first method uses the direct AMS results (i.e., ratios) to perform the blank correction, but does not consider the small variations that can be introduced within the samples during the carrier addition phase (e.g., Corbett et al., 2013; Nelson et al., 2014; Savi et al., 2016); specifically, it assumes that exactly the same amount of $^9$Be carrier was added to all samples and blanks. In the second approach (used in this paper), the AMS ratio is converted into $^{10}$Be atoms before performing the blank correction. By using the carrier weight and carrier concentration to calculate the precise amount of $^9$Be

added to each sample, this method allows obtaining the exact number of $^{10}Be$ atoms added from the carrier during the carrier addition phase ($^{10}Be_{car}$). After having calculated the number of $^{10}Be$ atoms contained in the sample ($^{10}Be_{S\_uncorr}$), the blank correction ($^{10}Be_{S\_corr}$) is performed by subtracting these two values (Eq. 1):

$$^{10}Be_{S\_corr} = {}^{10}Be_{S\_uncorr} - {}^{10}Be_{car} \tag{1}$$

5 It follows that when the number of $^{10}Be$ atoms in a sample is significantly greater than in the blanks, the value and variability of multiple blank measurements has little impact on the blank-corrected result. However, in the case of low $^{10}Be$ content in a sample, the blank correction may constitute a large subtraction, and variations among individual blanks become important.

Some $^{10}Be$ in the blank may originate from processes that would affect an entire batch of samples that are
10 processed together, such as the $^{10}Be$ contained in the carrier or in the stock chemicals used (see *supplementary material, text S5*). In these cases, a single blank per batch probably provides a good measure of the $^{10}Be$ contamination. Other sources, such as cross-contamination from poor laboratory practices or insufficient cleaning of reusable labware result in variable contamination among samples of the same batch and using a single blank per batch may thus be inadequate. Bierman et al. (2002) provide details of replicate blank measurements from 53
15 batches of samples processed at the cosmogenic-nuclide target preparation laboratories of the University of Vermont, where two blanks were processed per batch. The good agreement between these blank pairs suggests, at least in that laboratory, that inter-batch contamination is not an issue. However, this is a point that needs to be addressed for each laboratory and perhaps at the individual-user level.

20 **2.4 Determination limits**

**2.4.1 General statistical background**

Following the *International Union of Pure and Applied Chemistry* (IUPAC) definition, there is a minimum sample concentration that can be determined to be statistically different from an analytical blank in every analytical procedure (Long and Winefordner, 1983). The term "statistically different" implies the application of a statistical
25 approach that tries to answer the question "what is the lowest sample concentration that can be reliably distinguished from a blank?" (Currie, 1968; Long and Winefordner, 1983; McKillup and Darby Dyar, 2010; Schrivastava and Gupta, 2011; Bernal, 2014). This question can also be formulated as "what is the upper value of the blank distribution (i.e. the distribution of all available $^{10}Be$ blank measurements) that ensures a reliable distinction between blank and sample amounts?"

30 For cosmogenic studies, the previous question can be translated into the following *null hypothesis* (Fig. 1): "The number of $^{10}Be$ atoms in a given sample is not distinguishable from that within the blank(s)", which must be tested at a fixed confidence interval. Here, we give the example of a 1-tail test, because we are interested in defining an upper limit for the blank distribution. For variables that are normally distributed, the most common values used for confidence intervals, calculated according to equation (2) below, are $\pm k\sigma$, with $k = 1$, 2, or 3 (McKillup and
35 Darby Dyar, 2010), and $\sigma$ being the standard deviation of the distribution. By adopting this statistical approach, we accept the risk of committing an error, whose probability is determined by the chosen confidence interval. This error is known as

the "α-type", "Type I", or "False Positive", and occurs when a value is declared significantly different from a blank when it truly is not (McKillup and Darby Dyar, 2010), or in other words, when we conclude that there is an analyte when in reality there is none (e.g. a blank value falling outside the confidence interval, or a sample those [10]Be content is in large part derived from laboratory contamination) (Bernal, 2014). The probability of incurring in this error can be reduced by choosing a wider confidence interval. For example, a confidence interval of 95% is associated with a probability of an α-type error of 5%, whereas a confidence interval of 99% would reduce the probability of an α-type error to 1%. However, a wider confidence interval (i.e., a lower probability of an α-type error) might increase in turn the probability of a "β-type" error, known as "Type II error" or "False Negative", which occurs when a value is declared not significantly different from a blank when it truly is, or in other words, when we conclude that an analyte is not present, when it actually is (e.g., a sample falling in the range of the blanks' confidence interval) (Bernal, 2014). In any case, 
[revised manuscript text omitted]
. However, when dealing with very low $^{10}$Be content (i.e., in the same range on the blank values), we suggest not to go lower than this threshold, as choosing a lower confidence interval would increase the probability of incurring in a α-type error, thus increasing the risk of using a sample for a quantitative interpretation when in reality it may be representative of laboratory contamination.

Once calculated, the LOD represents the lowest number of $^{10}$Be atoms that can be distinguished from the blanks and, thus, can be used to limit the ages or the denudation rates shown by the dataset. Our results indicate that, after having established which samples are above the defined threshold, the type of blank correction and the representative value used for the blank subtraction (i.e., mean versus median value), may be important for samples with low $^{10}$Be content, whereas their importance becomes negligible for samples with more than $10^6$ $^{10}$Be atoms (Fig. 5). However, because the LOD and the results of the blank correction are strongly dependent on the chosen blank ensemble, the final decision on which approach to use is best evaluated case by case (Fig. 4).

**5.4 Implications for geomorphic applications**

With our blank and sample datasets, we demonstrated that depending on which approach is used for the calculation of the determination limits, the number of samples that can reliably be distinguished from blanks may vary strongly. In particular, considering only the LOD values for both subsets of blanks, we showed that the percentage of samples that cannot be reliably distinguished from the blanks varies between 5% and 37% (Fig. 3). This range shows that for geomorphic applications, the precision of the measurements and the cleanliness of the laboratory procedure can have a strong impact on the final interpretation of the data. For example, by using low $^{10}$Be/$^9$Be-ratio carrier and increasing the AMS current, Schaefer et al. (2009) were able to obtain very low and precise blank values (between 6,000 and 26,000 atoms with $1\sigma < 10\%$) and very precise sample measurements (between 70,000 and 1,000,000 atoms with $1\sigma < 10\%$). When applying the IUPAC recommendations to this dataset, the LOD is around 58,500 $^{10}$Be atoms, implying that all the samples are statistically distinguishable from the blanks and can be quantitatively used with a confidence of 99.9%. With these highly precise results, the authors measured exposure ages as young as $150 \pm 15$ years, dating moraines of the Little Ice Age.

In a case of similarly young boulders and rapidly denuding tributary catchments on an alluvial fan, Savi et al. (2016) showed that depending on which blank correction method is applied, the propagated error on the final exposure ages or denudation rates can vary up to 20%. Also, considering the $LOD_N$ as the lower threshold, these authors had 3 samples (about one fifth of a first set of processed samples) that could not be distinguished from the blanks, and could only be used to limit the exposure ages or denudation rates. However, by increasing the amount of sample processed during the chemical procedure for the following set of samples, and thus increasing the precision of the AMS measurements, all of the remaining samples could be statistically distinguished from the blanks with a confidence of 99.9%. These results were interpreted as quantifiable exposure ages as low as $50 \pm 8$ years, and denudation rates as fast as $13.8 \pm 2.6$ mm/yr. This study highlights how it may be possible to measure young exposure ages and fast denudation rates at a very high confidence level.

**6 Conclusions**

In this paper we have discussed the challenges related to the use of cosmogenic nuclide techniques in the case of low $^{10}$Be content, which are typically found in samples collected from rapidly eroding landscapes, young surfaces, or when a very limited amount of the target mineral is available for analysis. By adapting a method commonly used in analytical chemistry, we describe an approach to define a lower threshold above which samples with low $^{10}$Be content can be used in a quantitative way, accounting for laboratory cleanliness and contamination that may occur during the chemical procedure. This approach can be applied by the end-user of AMS measurements based on a number of different options for characterizing laboratory blanks.

In summary, in an ideal situation, the use of at least 20 blank values would guarantee a statistically reliable value for the limit of detection, LOD, which can be considered as the lowest threshold for the quantitative use of cosmogenic nuclide data. When samples with low $^{10}$Be content are expected, the user can process multiple blanks within a single sample batch. (e.g., Nelson et al., 2014). As an alternative, one can use a long-term value derived from several laboratory blanks processed over a limited time frame during which laboratory conditions are assumed to be nearly constant. When an acceptable number of blank values is available (i.e., minimum 20), finding the distribution that best describes the blank ensemble and using the 99.9$^{th}$ percentile of that distribution for the calculation of the LOD guarantees a more precise estimate of this threshold at the fixed confidence level.

Our analysis demonstrates the importance of producing low, precise, and reproducible blank measurements, as they reduce the value of the various determination limits, therefore increasing the number of samples that are distinguishable from laboratory blanks.can be used to quantitatively measure exposure ages or denudation rates. Particularly for samples with low $^{10}$Be content, to guarantee re-usable and comparable data it is important to report detailed information about the laboratory protocols, AMS raw results with related uncertainties, blank measurements (including both the measured ratios andas well as the amount and concentration of added carrier, in order to calculate the number of $^{10}$Be atoms in the blanks), the value and the procedure used to calculate the chosen determination limit, and the applied blank correction method.

**Author Contribution**

[revised manuscript text omitted]

Corbett, L., Bierman P., Graly J., Neumann T., and Rood D., 2013. Constraining landscape history and glacial erosivity using paired cosmogenic nuclides in Upernavik, northwest Greenland, Geol. Soc. Am. Bull., 125(9/10), 1539–1553.

Corbett, L.B., Bierman, P.R. and Rood, D.H., 2016. An approach for optimizing in situ cosmogenic [10]Be sample preparation. Quaternary Geochronology,  33, 24-34.

Corbett, L. B., Bierman P. R., Rood D. H., Caffee M. W., Lifton N. A., and Woodruff T. E., 2017. Cosmogenic [26]Al/[10]Be surface production ratio in Greenland, Geophys. Res. Lett., 44, 1350–1359, doi:10.1002/2016GL071276.

[revised manuscript text omitted]

Marrero S.M, PhillipsF.M., Borchers B., Lifton N., Aumer R., Balco G., 2016. Cosmogenic nuclide systematics and the CRONUScalc program. Quaternary Geochronology, 31, 160-187.

Matsuzaki H., Nakano C., Tsuchiya Y.S., Kato K., Maejima Y., Miyairi Y., Wakasa S., Aze T., 2007. Multinuclide AMS performances at MALT. Nuclear Instruments and Methods in Physics Research B, 259, 36–40.

Merchel, S., Bremser, W., Bourlès, D.L., Czeslik, U., Erzinger, J., Kummer, N. -a., Leanni, L., Merkel, B., Recknagel, S., Schaefer, U., 2013. Accuracy of $^9$Be-data and its influence on $^{10}$Be cosmogenic nuclide data. J. Radioanal. Nucl. Chem. 298, 1871–1878. doi:10.1007/s10967-013-2746-x

Mocak, J., Bond, A. M., Mitchell, S., Scollary, G., 1997. A statistical overview of standard (IUPAC and ACS) and new procedures for determining the limits of detection and quantification: application to voltammetric and stnpping techniques. Pure & Appl. Chern., 69(2), 297-328.

Nelson, A., Bierman P., Shakun J., and Rood D., 2014. Using in situ cosmogenic $^{10}$Be to identify the source of sediment leaving Greenland, Earth Surf. Processes Landforms, 39, 1087–1100.

[revised manuscript text omitted]

 *, Approximately 0.15 mg of $^9$Be carrier was added to each sample. The $^{10}$Be/$^9$Be ratios were measured in BeO targets by accelerator mass spectrometry (AMS) at the University of Cologne (Germany), which uses the KN01-6-2, KN01-5-3, and KN01-5-1 standards with nominal $^{10}$Be/$^9$Be values of 5.35 ×10-13, 6.32 ×10-12, and 2.71 ×10-11, respectively. Corrections followed the standard of Nishiizumi et al. [2007] with a $^{10}$Be half-life of 1.36 (± 0.07)×10$^6$ yrs.

**Table 3. Statistical parameters of the balnk distributions**

|  | Mean | Std.Error Mean* | Std.Dev. | Median | Std. Error Median** |
|---|---|---|---|---|---|
| **8-blank distribution** | 0.76 | 0.14 | 0.39 | 0.7 | 0.09 |
| **61-blank distribution** | 1.72 | 0.25 | 1.96 | 0.93 | 0.16 |

Values are multiplied for $10^4$ and are expressed in $^{10}$Be atoms; * for C=2000 and f=3 in Eq. (6); ** for C=2000 and f=5 in Eq. (6).

**Table 4**. Limit of Detection (LOD) and Limit of Quantification (LOQ) for the eight blank-ensemble (processed along with the samples) and the 61 GFZ long-term blanks. The subscript "N" after LOD and LOQ refers to the normal distribution whereas the "NB" refers to the negative-binomial distribution.

| | $LOD_N$ Atoms (x$10^4$) | $LOQ_N$ atoms (x$10^4$) | $LOD_{NB}$ atoms (x$10^4$) | $LOQ_{NB}$ atoms (x$10^4$) |
|---|---|---|---|---|
| Eight blanks (n=8) | 1.93 | 4.65 | 2.39 | 4.78 |
| Long-term blanks (n=61) | 7.59 | 21.31 | 11.45 | 30.20 |
| Confidence interval (%) | 99.9 | 100 | 99.9 | 100 |
| **Number of samples falling below the given limit** | | | | |
| Eight blanks (n=8) | 3 | 12 | 3 | 12 |
| Value in % | 5.3 | 21.1 | 5.3 | 21.1 |
| Long-term blanks (n=61) | 16 | 28 | 21 | 29 |
| Value in % | 28.1 | 49.1 | 36.8 | 50.9 |

a) High $^{10}$Be content

b) Low $^{10}$Be content

[revised manuscript text omitted]

---

## Short Comment (SC2) · 9 Aug 2017

Dear Dr. Granger,

Thank you for the helpful and constructive review. I have particularly appreciated the detailed work on the reviewer on commenting our paper and giving us suggestions on how to improve it, and the interest he showed for our work. Regarding his comments: I have now asked the AMS laboratory where we performed our analysis all the information mentioned in this revision. In particular, I have now information on the current used during the measurements, the 10Be counts for both standard and samples, the cross-talk contamination and B-interference. We will add all this information in the revised version of the manuscript and dedicate a chapter in commenting and discussing

them. Also, we will move the text in the supplementary material (section S5) into the main text, as suggested by the reviewer. For the statistics on the Poisson data, I will carefully consider all the literature suggested by the reviewer and adapt the manuscript accordingly. Thanks again for the support.

With best regards, Sara Savi